# The expression of virulence genes increases membrane permeability and sensitivity to envelope stress in *Salmonella* Typhimurium

**Malgorzata Sobota**[1], **Pilar Natalia Rodilla Ramirez**[1], **Alexander Cambré**[2],
**Andrea Rocker**[1], **Julien Mortier**[2], **Théo Gervais**[1,3], **Tiphaine Haas**[1], **Delphine Cornillet**[1],
**Dany Chauvin**[1,3], **Isabelle Hug**[1], **Thomas Julou**[1,3], **Abram Aertsen**[2]\*,
**Médéric Diard**[1]\*

**1** Biozentrum, University of Basel, Basel, Switzerland, **2** Department of Microbial and Molecular Systems, KU Leuven, Leuven, Belgium, **3** Swiss Institute of Bioinformatics, Basel, Switzerland

\* abram.aertsen@kuleuven.be (AA); mederic.diard@unibas.ch (MD)

## Abstract

Virulence gene expression can represent a substantial fitness cost to pathogenic bacteria. In the model entero-pathogen *Salmonella* Typhimurium (*S*.Tm), such cost favors emergence of attenuated variants during infections that harbor mutations in transcriptional activators of virulence genes (e.g., *hilD* and *hilC*). Therefore, understanding the cost of virulence and how it relates to virulence regulation could allow the identification and modulation of ecological factors to drive the evolution of *S*.Tm toward attenuation. In this study, investigations of membrane status and stress resistance demonstrate that the wild-type (WT) expression level of virulence factors embedded in the envelope increases membrane permeability and sensitizes *S*.Tm to membrane stress. This is independent from a previously described growth defect associated with virulence gene expression in *S*.Tm. Pretreating the bacteria with sublethal stress inhibited virulence expression and increased stress resistance. This trade-off between virulence and stress resistance could explain the repression of virulence expression in response to harsh environments in *S*.Tm. Moreover, we show that virulence-associated stress sensitivity is a burden during infection in mice, contributing to the inherent instability of *S*.Tm virulence. As most bacterial pathogens critically rely on deploying virulence factors in their membrane, our findings could have a broad impact toward the development of antivirulence strategies.

**Data Availability Statement:** All relevant data are within the paper and its Supporting Information files (S1 Data).

## Introduction

Bacteria constantly sense their environment and adapt to changes by modulating gene expression accordingly. In entero-pathogenic bacteria, physiological and environmental stimuli drive the expression of virulence genes and the outcome of the interaction with the host [1–3]. Virulence can maximize the fitness of pathogens via host exploitation [4,5]. On the other hand, virulence factors can be costly to produce, therefore requiring fine-tuned regulation to ensure balanced and timely expression [6]. However, how regulatory pathways have been

**Funding:** MD was supported by a Swiss National Foundation for Science professorship (PP00PP_176954). The group of AA was supported by a fellowship (1135116N, to AC) and a grant (G0C7118N) from the Research Foundation Flanders (FWO-Vlaanderen), and a postdoctoral fellowship (PDM/20/118, to JM) from the KU Leuven Research Fund. The funders had no role in study design, data collection and analysis, decision to publish, or preparation of the manuscript.

**Competing interests:** The authors have declared that no competing interests exist.

**Abbreviations:** cat, chloramphenicol acetyltrasferase; CCCP, carbonyl cyanide 3-chlorophenylhydrazone; CFU, colony-forming unit; DiOC2(3), 3,3′-diethyloxa-carbocyanine iodide; DiSC3(5), 3,3′-dipropylthiadicarbocyanine iodide; FDR, false discovery rate; HCD, higher-energy collisional dissociation; HS, heat shock; LB, Lysogeny broth; LCN2, Lipocalin-2; MIPS, monoisotopic precursor selection; NPN, N-phenyl-1-naphthylamine; p.i, postinfection; PI, propidium iodide; SDC, sodium deoxycholate; SOPF, specified opportunistic pathogen free; *S*.Tm, *Salmonella* Typhimurium; T3SS-1, Type 3 secretion system 1; TCEP, Tris(2-carboxyethyl)phosphin; TE, Tris-EDTA; WT, wild-type; GFP, Green Fluorescent Protein; OD, Optical Density; SCV, Salmonella Containing Vacuole.

selected for in order to tune virulence expression based on environmental stimuli remains to be understood. Here, we address this point in the notorious pathogen *Salmonella* Typhimurium (*S*.Tm).

*S*.Tm is a facultative intracellular entero-pathogen able to prosper in the intestinal lumen of a broad range of hosts [7]. The AraC-like transcription factor HilD is at the center of *Salmonella*'s regulatory network of virulence expression. HilD controls about 250 genes constituting the HilD regulon [8–10]. Expression of this regulon depends on various environmental parameters [2]. Upon entry into stationary phase in rich medium, HilD mediates the OFF to ON virulence switch in a subset of cells, which is reflected by the bimodal expression of reporter genes [11,12]. In the gut, the HilD regulon primes this subset of cells to swim toward, attach to and invade enterocytes [7,13,14]. The resulting intestinal innate immune response favors the growth and the transmission of *S*.Tm [5,15], thus improving *S*.Tm fitness at the population level [16].

However, the fitness cost of virulence expression at the single-cell level and the facultative intracellular lifestyle of *S*.Tm both strongly constrain the expression of the HilD regulon. The production of invasion factors controlled by HilD correlates with a 2-fold reduction of the growth rate [12]. Although such a substantial fitness cost could threaten the evolutionary stability of virulence in *S*.Tm [17], the heterogeneous "all or nothing" (ON/OFF) bimodal expression pattern [11] mitigates the cost by allowing only a fraction of *Salmonella* cells to engage in the costly virulence program [17]. Furthermore, independently from the fitness cost, once inside the *Salmonella*-containing vacuole (SCV), two-component systems silence invasion genes like the Type 3 secretion system 1 (T3SS-1), whereas the expression of a specific set of genes is stimulated to ensure the intracellular survival of *S*.Tm [18,19]. For instance, low pH and osmolarity are sensed by EnvZ/OmpR, which triggers T3SS-2 production via SsrA/B [20]. Moreover, PhoP/PhoQ represses invasion genes upon sensing low $Mg^{2+}$, low pH, and cationic antimicrobial peptides [21–23], which further ensures survival within host cells.

Nevertheless, a number of other signals modulate virulence expression in *S*.Tm outside host cells. Availability of carbon sources, amino acids, divalent cations, phosphate, and oxygen tilt the balance toward inhibition or activation of the HilD regulon (reviewed in [2,3]). Envelope stress, e.g., bile and heat, as well as misassembled outer membrane proteins, generally inhibits T3SS-1 production [24–27]. It is not clear why sensing environmental cues unrelated to the intracellular niche such as bile or excessive heat [26,27] should *per se* inhibit the expression of invasion genes. As such, the evolutionary causes of the repression of invasion factors outside of the SCVs remain to be determined. For this, we must understand what conditions the fact that some *S*.Tm cells should express invasion factors or not, how virulence gene expression intertwines with the general physiology of the bacteria, and to what extent this has shaped regulatory networks in *S*.Tm.

Since most invasion factors controlled by HilD are embedded within the envelope (i.e., T3SS-1, SPI-4 T1SS, flagella, and chemotactic receptors), we hypothesized that the expression of the HilD regulon could render *S*.Tm intrinsically more sensitive to envelope stress and that inhibiting the expression of HilD-controlled genes could correspondingly be an integral part of a general envelope stress response specific to *S*.Tm. This hypothesis was addressed by comparing outer membrane permeability, death rate, and HilD activity in response to envelope stresses in populations of *S*.Tm strains in which the expression of regulators and functions downstream of HilD was genetically tuned. This revealed a clear trade-off between membrane robustness and virulence expression in *S*.Tm. We show that this trade-off influences the selection for avirulent mutants within host, which could therefore inspire the design of novel anti-virulence strategies against *Salmonella*.

## Results

### Bimodal expression of the HilD regulon and proteomic analysis on sorted *S*.Tm cells

In this study, we used the chromosomal P*prgH*::*gfp* reporter (in which *gfp* expression is controlled by a copy of the promoter of the T3SS-1 *prg* operon) inserted in the locus *putPA* [11] as a proxy for the expression of the HilD regulon in *S*.Tm SL1344 (further referred to as wild type or WT). More specifically, the P*prgH*::*gfp* reporter is activated by HilA [28], itself tightly controlled by HilD [29], and does not interfere with T3SS-1 expression. The distribution of the *gfp* expression at late exponential phase in Lysogeny broth (LB) was clearly bimodal with approximately two-third of the population in the OFF state and one-third in the ON state (**S1 Fig**). As previously reported, the ON/OFF ratio results from the production of HilD, whose activity is controlled by the negative regulator HilE at the posttranslational level [12,17,30]. Accordingly, the proportion of ON cells was increased to about half of the population in the Δ*hilE* mutant. Note that the ON/OFF ratio in WT and its Δ*hilE* derivative varied between experiments, but the latter consistently yielded more ON cells than the WT (**Figs 1A, 2D, and 3A**). Deletion of *hilD*, on the other hand, locked the cells in the OFF state (**S1 Fig**), validating the P*prgH*::*gfp* reporter as a proper proxy for HilD activity.

To further confirm that the P*prgH*::*gfp* reporter is coexpressed with HilD-regulated genes, we compared the proteomic profile of WT *S*.Tm cells sorted according to the bimodal distribution of the green fluorescence signal (**S1 Fig, S1 Table**). The translation of SPI-1 T3SS-1 components and effectors, the SPI-4 T1SS, flagella, and chemotaxis systems was increased in GFP positive cells (**Table 1**), which was consistent with previously published transcriptomic data describing the HilD regulon [8–10]. This analysis validated the use of the P*prgH*::*gfp* fusion as reporter for HilD regulon expression at the single-cell level throughout this study.

### The expression of *hilD* increases the permeability of the outer membrane to a lipophilic compound

Several functions controlled by HilD are large protein complexes embedded in the envelope of *S*.Tm (T3SS-1, SPI-4 T1SS, flagella, and chemotactic receptor clusters (**S1 Fig, Table 1, S1 Table**), potentially affecting envelope integrity. In order to assess permeability of the outer membrane, we used N-phenyl-1-naphthylamine (NPN), a lipophilic dye that is weakly fluorescent in aqueous environments but becomes highly fluorescent in hydrophobic environments such as the inner leaflet of the outer membrane and the inner membrane [31]. In growth conditions triggering expression of the HilD regulon (i.e., late exponential phase in LB) (**Fig 1A**), the WT and the Δ*hilE* mutant accumulated significantly more NPN than the Δ*hilD* mutant (**Fig 1B**). In contrast, adding glucose to the media drastically reduced the expression of the HilD regulon [32] (**Fig 1A**) and NPN uptake (**Fig 1B**). As a control, polymyxin B, acting as a detergent, increased NPN uptake independently of glucose presence (**Fig 1C**). We tested a possible effect of *gfp* expression (used to monitor the HilD regulon induction) by constitutively expressing *gfp* from a plasmid (transcriptional fusion P*rpsM*::*gfp* in pM965) (**Fig 1A, S2 Fig**), which demonstrated that the presence of GFP did not affect NPN uptake or the measurement of the NPN fluorescent signal (**Fig 1B**).

We then evaluated the relative contribution of T3SS-1, SPI-4 T1SS, and flagella to the increased membrane permeability in *S*.Tm (**Fig 1D**). The full SPI-1 deletion (including *hilD*) phenocopied the Δ*hilD* mutant (*p* = 0.880), validating our previous observation. However, the deletion of the *iagB-invG* locus in SPI-1 (i.e., removing operons *iag*, *spt*, *sic*, *iac*, *sip*, *sic*, *spa*, and *inv*, but keeping transcriptional regulators *hilD*, *hilC*, *hilA*, and *invR* intact) or SPI-4

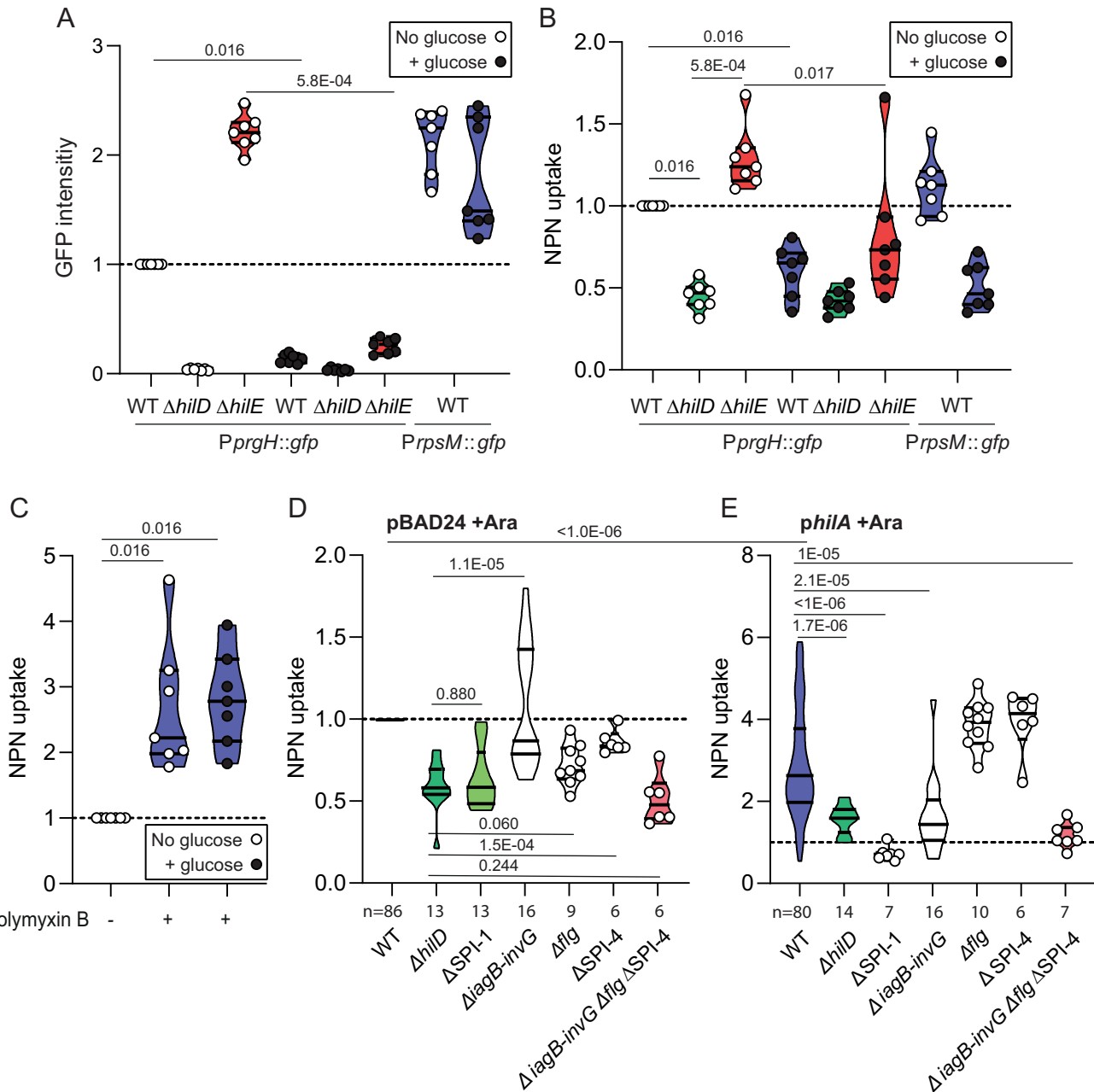

**Fig 1. Expression of the HilD regulon increases the permeability of the outer membrane to NPN.** GFP (from P*prgH*::*gfp*) **(A)** and NPN fluorescence **(B–E)** were measured from cells treated with 10 μM NPN and were divided by the optical density at 600 nm. Values of each repetition are normalized using a parallel experiment on the WT. When indicated, 0.1% of glucose was added to the broth in order to repress virulence expression (+ glucose, black dots). A control strain constitutively expressing GFP from pM965 carrying P*rpsM*::*gfp* (fluorescence distribution shown in **S2 Fig**) was used to control for the effect of GFP on the fluorescence readout from NPN uptake. A 2.5-μg/mL Polymyxin B treatment permeabilizing the membrane was used as positive control (C). (D, E) NPN uptake in WT, *ΔhilD*, *Δ*SPI-1, *ΔiagB-invG*, *Δflg*, *Δ*SPI-4, and triple mutant *ΔiagB-invG Δflg Δ*SPI-4 carrying either pBAD24 (D) or p*hilA* (E) and treated with 10 μM NPN. Fluorescence values were normalized to the reference WT pBAD24 grown in presence of 1 mM arabinose. Values obtained in the absence of arabinose are shown in **S2 Fig**. For comparisons against the WT, *p*-values were calculated using the raw data in paired Wilcoxon tests. For comparisons between mutants or conditions, *p*-values were calculated using the normalized dataset in unpaired Mann–Whitney tests. *p*-Values for comparisons discussed in the main text are indicated within the panels with bars marking the compared conditions. **S2 Table** shows *p*-values for comparisons between groups from panels **D** and **E**. Numbers below the x-axis (*n* = x) indicate the number of replicates when nonequal between conditions in a given experiment. *n* = 7 in panels **A–C**. Source data are provided as a source data file (**S1 Data**). NPN, N-phenyl-1-naphthylamine; WT, wild-type.

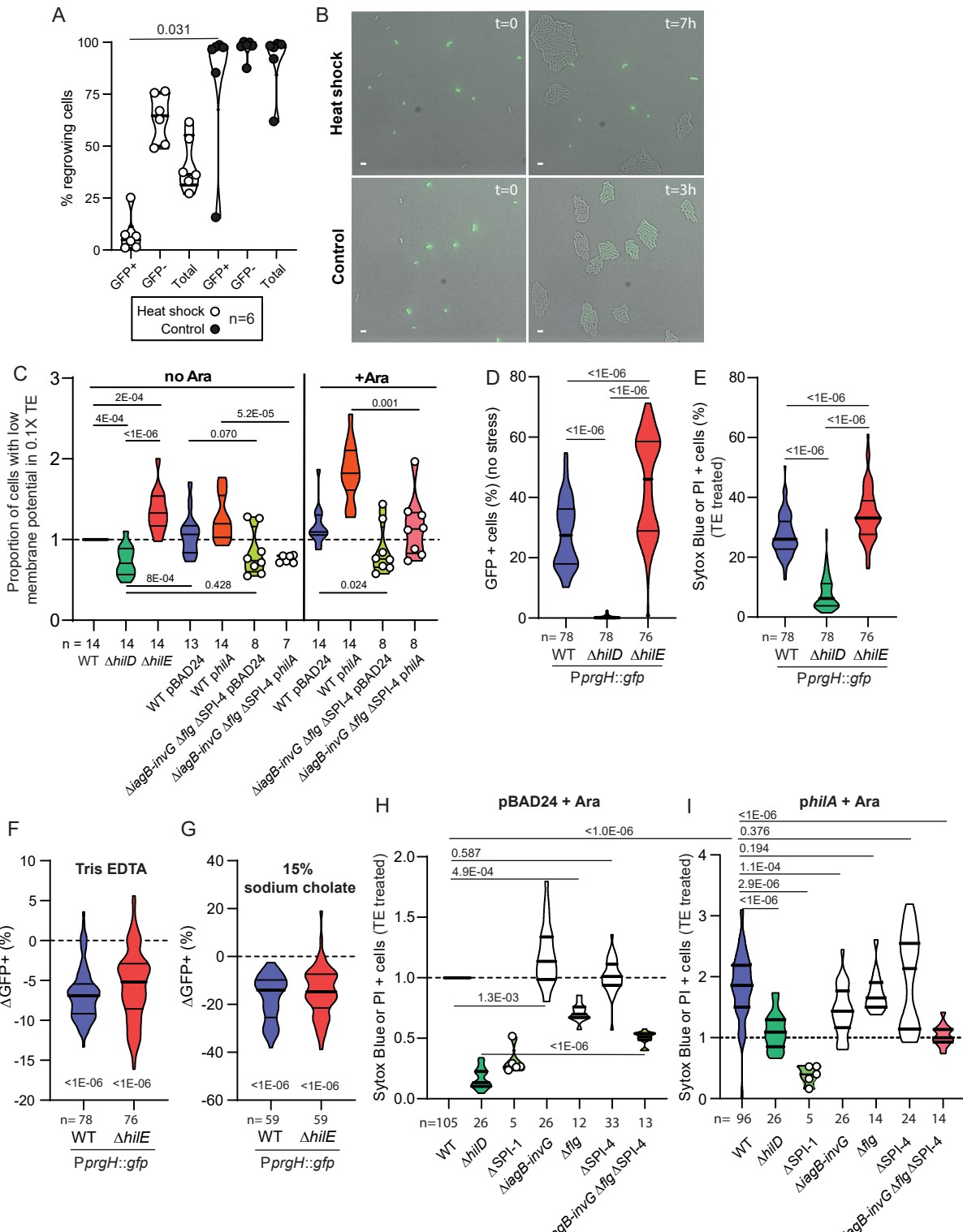

**Fig 2. Expression of the HilD regulon increases sensitivity to membrane stress. (A)** Time-lapse microscopy analysis of WT reporter strain (P*prgH*::*gfp*, GFP) after HS (51°C, 15 minutes) (white dots) and untreated control (black dots). Violin plots represent the fraction of cells able to form microcolonies among cells expressing the HilD regulon (GFP+), not expressing the HilD regulon (GFP–) and the total population. The *p*-value was calculated using a paired Wilcoxon test. **(B)** Representative pictures from time-lapse microscopy experiments. Cells in the upper panel were heat treated. Left picture shows cells at t = 0, and right picture shows cells after 7 hours. Cells in the lower panel are

untreated control. Left picture shows cells at t = 0, and right picture shows cells after 3 hours. Scale bar: 2 μm. **(C)** Cells grown in LB supplemented (+ Ara) or not (no Ara) with 1 mM L-arabinose to induce the overexpression of *hilA* from p*hilA* (derivative of pBAD24) were stained using 30 μm DiOC$_2$(3) in the presence of 10 mM Tris-1 mM EDTA (0.1X TE) and analyzed by flow cytometry. Unstained WT control and WT cells treated with 100 μm CCCP and 30 μm DiOC$_2$(3) were used to define the population of cells with low membrane potential in each sample (**S4 Fig**). The proportion of cells with low membrane potential was then normalized according to values obtained in parallel experiments using the WT. For comparisons against the WT, *p*-values were calculated on the raw data using paired Wilcoxon tests. For comparisons between mutants or conditions, *p*-values were calculated using the normalized dataset and unpaired Mann–Whitney tests. **(D)** Proportion of cells producing GFP (P*prgH*::*gfp*) from WT, *ΔhilD*, and *ΔhilE* strains, not stained by Sytox blue or PI (i.e., alive) in distilled water, measured by flow cytometry. **(E)** Frequency of cells stained with either Sytox blue or PI (i.e., dead) after treatment with 100 mM Tris-10 mM EDTA (TE treated) from WT, *ΔhilD*, and *ΔhilE* strains measured by flow cytometry. **(F, G)** Reduction of the GFP positive fraction (ΔGFP+) among WT or *ΔhilE* cells alive after TE treatment compared to distilled water control (F) or 15% sodium cholate compared to PBS control (G). Significance of the deviation of the median from 0 (dashed lines) estimated by Wilcoxon signed rank test (*p* < 1E-06). **(H, I)** Normalized frequency of cells stained with either Sytox blue or PI (i.e., dead) after TE treatment from *ΔhilD*, ΔSPI1, *ΔiagB-invG*, *Δflg*, ΔSPI-4, and triple mutant *ΔiagB-invG Δflg* ΔSPI-4 strains harboring pBAD24 (**H**) or p*hilA* (**I**). Data normalized to parallel WT pBAD24 controls. The cultures were supplemented with 1 mM arabinose (+ Ara) to induce *hilA* expression in the strains carrying p*hilA*. For comparisons against the WT, *p*-values were calculated using the raw data in paired Wilcoxon tests. For comparisons between mutants or conditions, *p*-values were calculated using the normalized dataset in unpaired Mann–Whitney tests. *p*-Values for comparisons discussed in the main text are indicated within the panels with bars marking the compared conditions. The **S5 Table** contains *p*-values for comparisons of data from panels **H** and **I**. Numbers below the x-axis indicate the number of replicates. Source data are provided as a source data file (**S1 Data**). DiOC2(3), 3,3′-diethyloxa-carbocyanine iodide; HS, heat shock; LB, Lysogeny broth; PI, propidium iodide; TE, Tris-EDTA; WT, wild-type.

individually had significantly less effect than the *ΔhilD* mutation (**Fig 1D**). Interestingly, deleting the *flgBCDEFGHIJ* operon (thereafter shortened *flg*) or combining deletions of *iagB-invG*, *flg* and SPI-4 phenocopied the *ΔhilD* mutant (*p* > 0.05) (**Fig 1D**).

Overproducing HilA, a transcriptional regulator of virulence (including T3SS-1 and SPI-4) controlled by HilD [33], led to a drastic increase in NPN uptake in the WT (*p* < 1E-06). In these experiments, strains carrying the empty vector pBAD24 (**Fig 1D**) were compared with strains overexpressing *hilA* from the pBAD24 derivative p*hilA*, both growing in the presence of the arabinose inducer (**Fig 1E**).

Upon *hilA* overexpression, compared to WT, the *iagB-invG* deletion reduced membrane permeability (*p* = 2.1E-05) as well as the triple deletion *iagB-invG flg* SPI-4 (p = 1E-05). However, deleting the *flg* operon (potentially repressed by HilA via inhibition of *flhD* [33]) or SPI-4 did not reduce membrane permeability.

A reproducible pattern of NPN uptake was observed when comparing various mutants in which *hilA* was not overexpressed (pBAD24 carrying strains or in absence of arabinose), confirming that the flagella were the most important contributors to membrane permeability (**Fig 1D**, **S2 Fig**). Unexpectedly, the impact of the T3SS-1 was marginal at WT expression level and production of the SPI-4 T1SS remained relatively neutral in all tested conditions. **S2 Table** gathers statistical analysis results from this dataset.

## The expression of *hilD* reduces resistance to outer membrane disrupting treatments

In general, permeability to NPN is increased by treatments that disrupt the membrane of gram-negative bacteria like polymyxin B (**Fig 1C**) [34], indolicidin [35], and aminoglycosides acting as divalent cation binding sites on the outer membrane [36]. Increased NPN uptake in *hilD* expressing populations of *S*.Tm (**Fig 1B**) suggested that the membrane of HilD ON cells could be inherently disrupted, thus making these cells more sensitive to stress targeting the membrane.

Since heat provokes disruption of the outer membrane [37], we first measured survival of *S*. Tm exposed to a mild heat shock (HS, 51°C, 15 minutes) at the single-cell level with time-lapse fluorescence microscopy. The cells were observed for 16 hours posttreatment. The vast majority of cells expressing the HilD regulon (i.e., GFP positive cells) was unable to resume growth

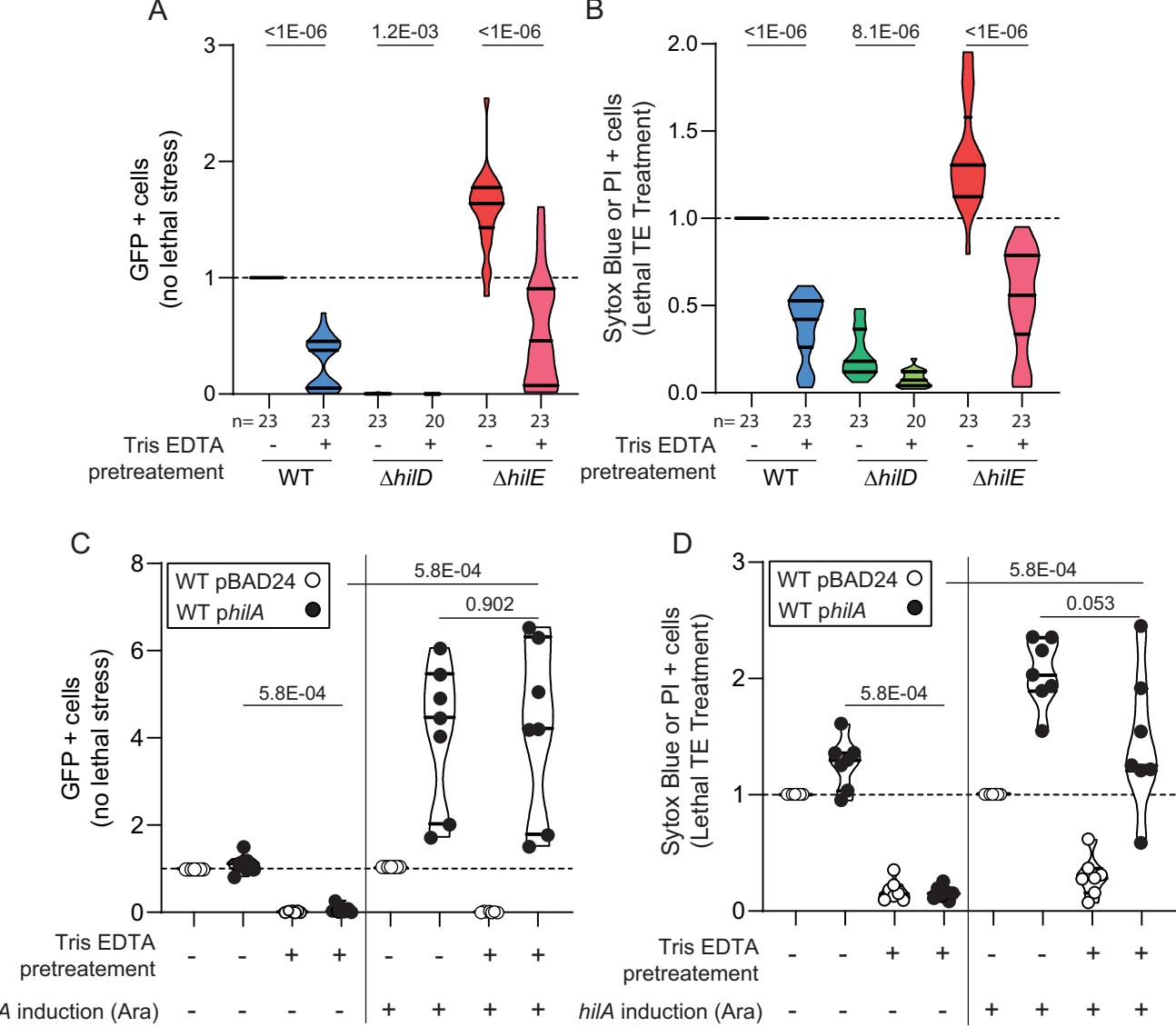

**Fig 3. Sublethal stress inhibits expression of virulence and increases resistance against lethal TE exposure.** Flow cytometry analysis. **(A)** Proportion of GFP expressing cells (P*prgH*::*gfp*) from WT, *ΔhilD*, and *ΔhilE* strains not stained by Sytox blue or PI (i.e., alive) in absence of lethal treatment (distilled water control). Data normalized to parallel experiments with the WT strain. When indicated, 4 mM Tris–0.4 mM EDTA was added to the broth as sublethal pretreatment. **(B)** Frequency of cells stained with either Sytox blue or PI (i.e., dead) after treatment with 100 mM Tris-10 mM EDTA (lethal TE treatment) in WT, *ΔhilD*, and *ΔhilE* strains. Data normalized to parallel experiments with the WT strain. When indicated, 4 mM Tris–0.4 mM EDTA was added to the broth. **(C)** Proportion of GFP expressing cells, unstained by Sytox blue or PI, in distilled water. WT strain carrying the empty vector pBAD24 or the p*hilA* plasmid allowing for overexpression of *hilA*. When indicated, 1 mM arabinose (Ara) and/or 4 mM Tris 0.4 mM EDTA were added to the broth. **(D)** Frequency of cells stained with either Sytox blue or PI after treatment with 100 mM Tris-10 mM EDTA. When indicated, 1 mM arabinose and/or 4 mM Tris–0.4mM EDTA were supplemented in the medium. (C, D) Data normalized using WT pBAD24 (−Ara or +Ara) as reference, *n* = 7 replicates. For comparisons against the WT, *p*-values were calculated using the raw data in paired Wilcoxon tests. For comparisons between mutants or conditions, *p*-values were calculated using data normalized using corresponding WT or WT pBAD24 as reference in unpaired Mann–Whitney tests. Numbers below the x-axis indicate the number of replicates. Numbers within the graphs are *p*-values for comparisons discussed in the main text and bars below these numbers indicate the compared groups. Source data are provided as a source data file (**S1 Data**). PI, propidium iodide; TE, Tris-EDTA; WT, wild-type.

(**Fig 2A and 2B** HS upper panels, and **S3 Fig**), while the rest of the population (i.e., GFP negative cells) regrew after a lag period. Untreated cells were able to grow normally regardless of their *hilD* expression state, with ON cells switching OFF and diluting the GFP (**Fig 2B** control lower panels, **S3 Fig**).

**Table 1. Membrane-embedded multiprotein complexes coproduced with GFP from the P*prgH*::*gfp* fusion.**

| Functions | Proteins | log2 fold increases ranging from |
|---|---|---|
| Flagellum and chemotaxis system | FliZ, FliB, FliC, FliS, FliL, FljB, MotA, MotB, Trg, CheB, CheM, CheW, CheA, SL1344_3112, SL1344_3189, Aer, and Tcp | 0.63 to 2.20 |
| T3SS-1 components, regulators, and effectors (SPI-1) | AvrA, OrgB, OrgA, PrgK, PrgJ, PrgI, PrgH, HilD, HilA, SptP, IacP, SipA, SipD, SipC, SipB, SicA, SpaS, SpaP, SpaO, InvJ, InvC, InvB, InvA, InvE, InvG, InvF, and InvH | 1.60 to 5.47 |
| SPI-4 T1SS | SiiA, SiiB, SiiC, SiiD, SiiE, and SiiF | 2.29 to 6.22 |

*S*.Tm cells expressing *gfp* controlled by P*prgH* produce significantly more flagella, T3SS-1, and SPI-4 T1SS proteins. These membrane-embedded multiprotein complexes are coproduced with additional proteins listed in S1 Table constituting the HilD regulon.

To determine if membrane permeability correlated with higher sensitivity to stress in the ON cells, we analyzed HS sensitivity of the triple Δ*iagB-invG* Δ*flg* ΔSPI-4 mutant, which has already proved less permeable to NPN (**Fig 1**). This mutant produced an amount of *hilD* expressing cells comparable to the WT (**S3 Fig**), with similarly reduced growth rate compared to OFF cells (**S3 Fig**). However, a higher proportion of these ON cells was able to resume growth after HS than in WT or the Δ*hilE* mutant (**S3 Fig**), suggesting that membrane-localized virulence factors increase both membrane permeability and stress sensitivity.

We then measured membrane potential in cells exposed to 10 mM Tris-1 mM EDTA (0.1X Tris-EDTA [TE]), which destabilizes the lipopolysaccharide of gram-negative bacteria [38] and allows entry of the dye 3,3′-diethyloxa-carbocyanine iodide (DiOC$_2$(3)) (**Fig 2C**) [39]. In the presence of 0.1X TE, membrane potential leads to accumulation of DiOC$_2$(3) to the point of dye aggregation shifting its fluorescence from green to red. Here, we used flow cytometry because it measures fluorescence in a high number of cells and allows testing multiple conditions in parallel. We followed the gating strategy described in **S4 Fig** to estimate the proportion of cells with low membrane potential (green only) among the population of stained cells (red and green) in WT and mutant strains. Exposure to the proton ionophore carbonyl cyanide 3-chlorophenylhydrazone (CCCP) was used as control condition in which the membrane potential is abolished and most DiOC$_2$(3) stained cells remained green (depolarized membrane). Unstained cells and cells exposed to CCCP and DiOC$_2$(3) served as references (**S4 Fig**). Representative images are provided in **S4 Fig**.

Again, a pattern consistent with NPN uptake emerged from these experiments. The WT strain and the Δ*hilE* mutant showed significantly higher proportion of cells with low membrane potential compared to the Δ*hilD* mutant. Overexpression of *hilA* in the presence of arabinose increased the proportion of cells with low membrane potential. This was rescued in the Δ*iagB-invG* Δ*flg* ΔSPI-4 triple mutant (**Fig 2C**). In the context of endogenous *hilA* expression, the triple mutant Δ*iagB-invG* Δ*flg* ΔSPI-4 phenocopied the Δ*hilD* mutant ($p = 0.428$).

Membrane potential in absence of stress was measured in control experiments using 3,3′-dipropylthiadicarbocyanine iodide (DiSC$_3$(5)) (**S5 Fig**), a red fluorescent hydrophobic probe accumulates in the polarized membrane of cells [40]. This assay was compatible with the readout for expression of the HilD regulon with the reporter P*prgH*::*gfp*. We observed no difference in DiSC$_3$(5) staining in *hilD* ON (GFP+) versus OFF cells (GFP−) (**S5 Fig**), suggesting that the membrane of *hilD* expressing cells were not inherently depolarized. Reduction of membrane potential observed with DiOC$_2$(3) was therefore due to exposure to 10 mM Tris-1 mM EDTA

further destabilizing the outer-membrane, especially when *S*.Tm expressed the HilD regulon (**Fig 2C**).

This observation led us to evaluate the susceptibility of the ON cells to a more severe 100 mM Tris-10 mM EDTA treatment (TE treatment). We used two complementary approaches, flow cytometry and microscopy, to quantify the proportion of dead cells detectable after TE treatment and the fraction of cells expressing the HilD regulon (GFP+ cells) among the survivors. We used two dyes to stain the dead cells: propidium iodide (PI) or Sytox blue. **Fig 2D** shows the proportion of GFP+ cells exposed to distilled water used as solvent for TE. After TE treatment, we observed a clear increase in the proportion of dead cells in WT and the *ΔhilE* mutant compared to the *ΔhilD* mutant with both dyes (**Fig 2E**). Although Sytox blue had the tendency to stain slightly more cells than PI (**S6 Fig**), we judged the overlap sufficient to pool both staining results in every dataset. The proportion of ON cells decreased in the surviving populations compared to control (**Fig 2F**). The timing of the experiment (30' of treatment before cytometry analysis) was too short to allow ON cells to switch OFF and to dilute the GFP by cell division. The constitutive expression of *gfp* from the promoter P*rpsM* did not alter the overall pattern of stress sensitivity when comparing WT, *ΔhilD*, and *ΔhilE* strains (**S6 Fig**). For unclear reasons, WT and *ΔhilD* with P*rpsM*::*gfp* were slightly less sensitive than their P*prgH*::*gfp* counterparts. This nevertheless suggested that expressing *gfp* was not increasing stress sensitivity *per se*. We also ruled out the contribution of nonfluorescent and unstained debris formed during treatment by quantifying the fraction of nonfluorescent events from stressed cells constitutively expressing *gfp* (**S6 Fig**). Moreover, a treatment with 15% sodium cholate, a natural detergent present in bile, was even more potent at reducing the proportion of ON cells in WT (−14% versus −6.8% with TE) and the *ΔhilE* mutant (−14.7% versus −5.2% with TE) (**Fig 2G**). The effect of lethal TE treatment was further confirmed by live monitoring of cells in a microfluidic device (**S1 Movie**). In fact, mainly the ON cells from the WT reporter strain exposed to TE died and became stained by PI (red) concomitantly with losing their GFP, while most OFF cells remained apparently intact (quantification in **S7 Fig**). Imaging of PI and Sytox blue staining upon TE treatment is presented in **S6 Fig**.

Additional microscopic analysis of WT reporter bacteria after TE treatment confirmed that, although ca. 70% of the cells were able to resume growth (**S8 Fig**), the fraction of GFP + cells among the regrowing cells was significantly reduced (17% less GFP+ cells compared to untreated control (**S8 Fig**)). This indicated that the TE treatment killed a significant amount of bacteria, with a higher probability for the ON cells to die. Flow cytometry and microscopy showed comparable results with 30% of WT cells stained by PI or Sytox blue (**Fig 2E**) or not regrowing on the agar pad after TE treatment (**S8 Fig**). Based on this overall death rate and parameters extracted from cytometry experiments (**S3 Table**), we estimated a 44% death rate for the ON cells, about 2.3 times higher than the death rate of the OFF cells (19%) from WT *S*.Tm (Formula: Death rate = 100 −((% final × % total survivors)/% initial); median values, **S3 Table**). These values were comparable in the *ΔhilE* mutant (41% and 28% respective death rate for ON and OFF cells). However, the death rate of the locked OFF *ΔhilD* mutant (6.2%) was strikingly lower than for WT or *ΔhilE* mutant OFF cells. This could be attributable to the method used to discriminate between the ON and OFF cells from WT and *ΔhilE* mutant strains based on a fluorescent intensity cutoff when using flow cytometry or microscopy. Meaning that, below this detection cutoff, cells in which the HilD regulon is not fully repressed could bias the overall death rate of the so-called "OFF" subpopulations in WT and *ΔhilE*. Counting colony-forming units (CFUs) posttreatment confirmed the overall death rate being higher in WT and the *ΔhilE* mutant compared to the *ΔhilD* mutant (**S4 Table**). These data also suggested that Sytox blue or PI staining and counting the regrowing cells under the microscope could underestimate the fraction of cells affected by TE treatment and potentially lysing

(**S4 Table**). Alternatively, plating might be an extra stress that kills cells after TE treatment which would otherwise form microcolonies under the microscope or would not be stained by PI or Sytox blue.

As observed for NPN uptake, the deletion of *iagB-invG* or SPI-4 alone did not change sensitivity to TE. However, the individual deletion of the *flg* operon significantly increased resistance to TE compared to WT ($p$ = 4.9E-04) (**Fig 2H**).

Although the cumulative deletions of *iagB-invG*, *flg* and SPI-4 increased resistance to TE treatment, they did not fully phenocopy the resistance of the Δ*hilD* mutant (**Fig 2H**) ($p < $ 1E-06). This could be due to the higher sensitivity of this assay compared to NPN uptake measurements in batch cultures. It also suggests that other functions controlled by HilD could play a role in increasing stress sensitivity in the ON cells.

The overexpression of *hilA* (p*hilA* + arabinose) drastically enhanced sensitivity to TE ($p < $ 1E-06) (**Fig 2I**). Under these conditions, the full SPI-1 deletion ($p$ = 2.9E-06) and, to a lesser extent, the deletion of *hilD* alone ($p < $ 1E-06), of *iagB-invG* ($p$ = 1.1E-04), and of *iagB-invG flg* SPI-4 ($p < $ 1E-06) restored some resistance. Individual deletions of *flg* or SPI-4 had no effect ($p > $ 0.05). Controls in the absence of arabinose reproduce the pattern observed for strains carrying pBAD24 in the presence of arabinose (**S9 Fig**). **S5 Table** gathers statistical analysis results from this dataset.

## Exposure to sublethal Tris–EDTA concentration inhibits expression of virulence and increases resistance to lethal stress

Given that the expression of the HilD regulon sensitizes *S*.Tm to heat, sodium cholate and TE exposure, and previous observations that bile or HS inhibit the expression of T3SS-1 [24–27], we hypothesized that exposure to sublethal stress might result in shutting down the HilD regulon, which should protect *S*.Tm against lethal stress.

To address this, we grew the bacteria in LB medium containing TE at sublethal concentration (4 mM Tris–0.4 mM EDTA, 0.04X TE) before exposure to a lethal dose (100 mM Tris-10 mM EDTA). As expected, the proportion of ON cells was strikingly reduced when *S*.Tm grew in the presence of TE at sublethal concentration (**Fig 3A**), and survival was increased when the bacteria were then exposed to lethal dose (**Fig 3B**). Overexpression of *hilA* restored P*prgH*::*gfp* expression (**Fig 3C**, **S10 Fig**) and, with it, the high death rate in sublethal TE pretreated cells exposed to lethal TE concentration (**Fig 3D**). This suggested that the expression of genes downstream of *hilA* increased stress sensitivity even if a protective stress response is triggered by sublethal stress. Shutting down the expression of the HilD regulon upon sublethal stress increases the chance of surviving harsher environments that would otherwise be lethal if cells would remain ON.

## HilD-mediated stress sensitivity is costly during symptomatic infections

In contrast to a Δ*hilD* mutant, the *iagB-invG flg* SPI-4 triple deletion restored stress resistance but did not prevent the growth defect associated with HilD expression in the ON cells (**S3 Fig**). This allowed us to determine whether HilD-mediated stress sensitivity was in itself a significant burden during infection in mice (**Fig 4**, **S12 Fig**). For this, we compared the outcome of competitions between the WT (yielding ON cells displaying both slow growth and stress sensitivity) and the Δ*hilD* mutant (not yielding ON cells) (competition 1), with competitions between the triple mutant (yielding ON cells displaying slow growth) and the Δ*hilD* triple mutant (not yielding ON cells) (competition 2). To ensure comparable conditions during infections, a fully virulent helper strain (untagged WT) was added to the tagged competitors (**Fig 4A**). The role of the helper strain was to trigger inflammation that would otherwise not be

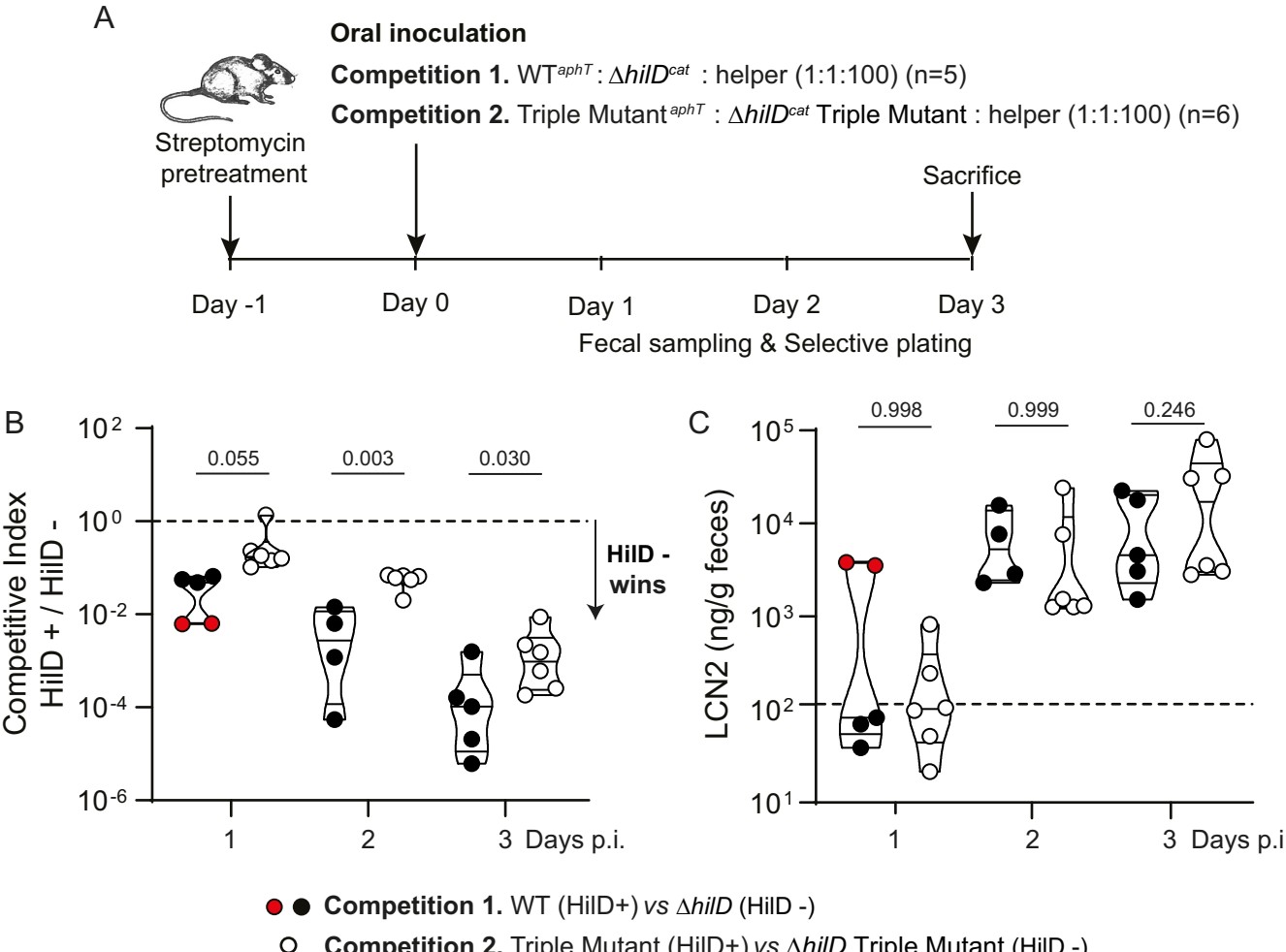

**Fig 4. Virulence-associated stress sensitivity is costly during infection. (A)** Schematic diagram of the experimental infection setup. Two competitions were performed in the presence of a virulent helper strain triggering inflammation in C57BL/6J mice pretreated with streptomycin. Competition 1 featured *S*.Tm WT (yielding ON cells displaying both slow growth and stress sensitivity) and the *ΔhilD* mutant (not yielding ON cells). Competition 2 featured the *ΔiagB-invG* Δ*flg* ΔSPI-4 triple mutant (yielding ON cells displaying slow growth) and the *ΔhilD* triple mutant (not yielding ON cells). **(B)** Competitive indices calculated from the relative proportions of each competitor in fecal samples. Dots correspond to individual mice at a given time point. Filled dots (black or red depending on the inflammation at day 1 p.i.): competition 1 (WT versus *ΔhilD*); empty dots: competition 2 (Triple mutant versus *ΔhilD* triple mutant). The dashed line represents a 1:1 ratio between HilD+ and HilD− competitors. Values below this line indicate that the HilD− strain outgrew the HilD+ strain. **(C)** Intestinal inflammation estimated by measuring LCN2 concentration in the feces. Concentrations above $10^2$ ng/g are usually detected in feces from inflamed gut (dashed line). Red dots indicate mice particularly inflamed during competition 1 at day 1 p.i. Two independent inocula per condition. The *p*-values indicated within graphs were generated with one-way ANOVA and corrected by a Sidak multiple comparisons test; bars below *p*-values mark the compared groups. Source data are provided as a source data file (**S1 Data**). LCN2, Lipocalin 2; *S*.Tm, *Salmonella* Typhimurium; WT, wild-type.

triggered by strains harboring deletions inactivating T3SS-1 and motility. The helper was added at a 100-fold excess in the inocula to prevent the *ΔhilD* mutants from outgrowing it.

The triple mutant was outgrown by its *ΔhilD* derivative, but at a significantly slower pace than the single *ΔhilD* mutant outgrowing the WT (**Fig 4B**, **S11 Fig**). This suggested that, in addition to the growth defect, increased stress sensitivity counts as a substantial burden in WT *S*.Tm cells expressing virulence *in vivo*. Differences between competitive indexes were significant at day 2 and 3 post-infection (p.i.), correlating with the onset of inflammation quantified by measuring the amount of Lipocalin 2 (LCN2) in the feces (**Fig 4C**) [41]. Along those lines,

the Δ*hilD* mutant outgrew the WT faster at day 1 p.i. in 2 mice particularly inflamed at this time point (marked with red dots in **Fig 4B and 4C**).

To further address the role of inflammation, we suppressed it by performing competitions between HilD+ versus HilD− strains using an avirulent genetic background unable to produce functional T3SS-1 (Δ*invG*) and T3SS-2 (Δ*ssaV*). Because deleting *iagB-invG* (**Fig 2**) or *invG* alone (**S12 Fig**) was not sufficient to reduce stress sensitivity in the WT, and because the T3SS-2 is mainly produced in the SCV, we were able to use Δ*invG* Δ*ssaV* avirulent mutant backgrounds without interfering significantly with stress resistance (experimental setting **S12 Fig**). In the absence of inflammation, the fitness disadvantage of HilD+ strains relative to Δ*hilD* mutants was not affected by the ability to express or not the full HilD regulon (competitions 3 and 4, **S12 Fig**). The pace at which HilD− outgrew HilD+ strains was rather slow compared to experiments performed with virulent strains (**Fig 4**). This may be because the inflammation kills part of the *Salmonella* population [42], therefore accelerating outgrowth of HilD− strains during regrowth. The whole population size of *Salmonella* was decreasing by day 3 p.i. because of competitive exclusion by the regrowing intestinal flora in the absence of inflammation [5]. Competitions 3 and 4 show that only the growth defect influences the outcome of competitions in the absence of inflammation when stress is mild and, with it, the burden of expressing the HilD regulon. Inflammation instead negatively impacts the ON cells weakened by the expression of virulence factors embedded in the membrane. These results demonstrate that stress upon inflammation could contribute to select for avirulent *S*.Tm mutants that naturally emerge during infection [17].

## Discussion

In *S*.Tm, the cost of virulence gene expression drives within-host emergence of attenuated mutants harboring loss-of-function mutations in positive transcriptional regulators of virulence (e.g., *hilD* and *hilC* [17,43]). Such attenuated mutants consistently emerge during chronic infections in mice and loss-of-function mutations in *hilD* were detected in a large collection of sequenced natural isolates at a frequency suggesting positive selection [44]. The genetic instability of virulence in *S*.Tm could be exploited to fight against this pathogen, which is becoming increasingly resistant to antibiotics [45]. However, the tight regulation of bimodal expression of virulence impairs fixation of attenuated mutants during within-host growth, ensuring transmission of the virulent genotype [17]. Understanding the cost of virulence, and how it relates to expression regulation, could allow the identification and the modulation of ecological factors in order to drive the evolution of *S*.Tm toward attenuation.

Until now, the only identified cost of virulence in *S*.Tm was a 2-fold reduction of the growth rate in cells expressing the HilD regulon [12]. The molecular mechanism underlying this growth defect is still unclear. Here, we discovered that the expression of *hilD* increased membrane permeability to the hydrophobic compound NPN (**Fig 1**) and sensitized *S*.Tm to stresses that disrupt the envelope (i.e., short exposure to mild heat, TE, and sodium cholate (**Fig 2**)). To identify the functions involved, we compared mutants lacking different operons upregulated by HilD coding for the T3SS-1, the flagella, and the SPI-4 T1SS. At WT expression level, the deletion of several T3SS-1 components (*iagB* to *invG*) did not significantly change membrane permeability or stress resistance (**Figs 1D and 2H**). Overexpressing *hilA*, which directly controls T3SS-1 gene expression, increased membrane permeability and sensitivity to stress. In this case, the deletion of *iagB-invG* did reduce membrane permeability to NPN and stress sensitivity (**Figs 1E and 2I**). This might relate to the fact that mislocalized T3SS components, like overexpressed secretin, can disrupt the membrane, as reported in *Yesinia enterocolitica* and *Escherichia coli* mutants that lack functional phage-shock proteins [46,47], as well as in

*Pseudomonas aeruginosa* [48]. However, we found that the flagella played a significant role, increasing stress sensitivity in the WT, in accordance with the observation that producing flagella increases the intrinsic death rate in *E. coli* MG1655, a strain that does not possess a *bona fide* T3SS [49]. Cumulating deletions showed that multiple functions controlled by HilD act synergistically to increase sensitivity to membrane stress.

Surprisingly, the cumulated deletions of most of the T3SS-1 component coding genes (*iagB-invG*), the *flg* operon and SPI-4, while increasing stress resistance, did not prevent the reduction in growth rate in the *hilD* expressing cells (**S3B Fig**). From this, we conclude that stress sensitivity and growth defect are actually two independent costs of virulence, the second most likely related to the production of T3SS-1 effectors (SopE, SopE2, SopB, SipA, SptP, SopA, SpvB, and SpvC), which, when removed in addition to the T3SS-1 translocon (*SipB*, *SipC*, and *SipD*), partially restore the growth rate of *S*.Tm [12]. Our *in vivo* competition results (**Fig 4**) also highlight that this independent cost of stress sensitivity is physiologically relevant, as it sensitizes cells expressing the HilD virulence program to the harsh environment in the inflamed gut [42].

The relative contribution of envelope destabilization by multiprotein complexes (T3SS-1 and flagella) and of the energetic burden of secretion and motility to the fitness cost of virulence is difficult to assess as these processes inevitably intertwine. Nevertheless, the comparative proteomic analysis of untreated ON versus OFF cells did not reveal any significant extracytoplasmic stress response in the ON cells, such as the induction of the RpoE or Cpx regulons (**S1 Table**) [50]. Therefore, the bacteria expressing the HilD regulon did not seem more intrinsically stressed than the OFF cells. As previously shown in *Pseudomonas putida* expressing the flagella [51], an energetic cost of virulence can impair the ability to cope with certain stresses. However, membrane potential was comparable between *hilD* ON and OFF *S*.Tm cells.

The repression of the HilD regulon by sublethal exposure to TE was in accordance with known inhibition of T3SS-1 production in *S*.Tm exposed to envelope stress (e.g., heat, bile, or cationic antimicrobial peptides [25–27]) and deprivation of divalent cations sensed by the two-component system PhoPQ [21]. The artificial overexpression of *hilA* showed that decoupling stress response from inhibition of virulence expression was detrimental to *S*.Tm in harsh environment. To the best of our knowledge, this is the first dataset demonstrating a link between regulation of virulence expression and stress response in *S*.Tm driven by the fitness cost of virulence. We propose that such trade-off between virulence and membrane robustness in *S*.Tm shaped the regulation of virulence expression by integrating it into the general stress response. Moreover, since many virulence determinants are typically deployed at the level of the cell envelope, this trade-off and regulatory integration might constitute a more general paradigm relevant for other bacterial pathogens.

To conclude, we demonstrate that the cost of virulence in *S*.Tm is pleiotropic but strongly envelope-related and that such virulence costs can be mitigated by down-regulation concomitant to stress response. Our work reveals that impaired membrane robustness constitutes in itself a significant selective force that can promote the rise of more resistant avirulent mutants within infected hosts (**Fig 4**) [17,43]. This occurs despite the fact that functions less expressed in these mutants, such as flagella and chemotaxis systems, favor growth in the inflamed gut [52]. Therefore, biological stress acting more specifically on the *hilD* expressing cells with impaired membrane robustness could be identified and harnessed for the design of novel antivirulence strategies. Many pathogens rely on similar costly features to ensure pathogenesis (e.g., entero-pathogenic *E. coli*, and *P. aeruginosa*). Hence, our discovery could have broad impact for the development of strategies aiming at fighting pathogens by interfering with the evolutionary stability of virulence.

## Materials and methods

### Strains and media

All the strains and plasmids used in this study are listed in **S6 Table**. *Salmonella enterica* serovar Typhimurium SB300 (SL1344) [53] and derivatives were cultivated at 37˚C using LB liquid or solid media (Difco United Kingdom). Antibiotic selection was performed with 100 μg/ml ampicillin (Sigma), 6 μg/ml chloramphenicol (Sigma Saint-Louis, MO), and 50 μg/ml kanamycin (Sigma) when needed. Subcultures in late exponential phase (induction of the HilD regulon) were prepared by diluting overnight cultures 1/100 in 2 mL LB and 4-hour incubation at 37˚C with shaking in absence of antibiotic. When stated, media were supplemented with 0.1% w/v D-glucose (Sigma) to inhibit the expression of the HilD regulon. For induction of *hilA* under the inducible P*ara* promoter in p*hilA* [54], a plasmid deriving from pBAD24 [55], media were supplemented with 1 mM L-arabinose (Sigma). Constitutive *gfp* expression was obtained from P*rpsM*::*gfp* cloned in pM965 [13]. Sublethal pretreatment of cells was performed using a mixture of 4 mM Tris (Sigma) and 0.4 mM EDTA (Sigma) added to the subculture broth. Spent LB medium used to trigger expression of the HilD regulon in microfluidic settings was prepared by sterile filtration of the supernatant from centrifuged late exponential phase WT cultures (OD$_{600}$ = 0.8).

### Mutant constructions

*Salmonella* mutants used in this study were constructed by homologous recombination using the λ-Red gene replacement system as described in [56]. To select for recombinants, the chloramphenicol acetyltrasferase (*cat*) gene was amplified from the pKD3 template plasmid using primers containing 40-bp region homologous to flanking regions of the target gene in the chromosome of *Salmonella*. The PCR product was transformed by electroporation into the recipient strain harboring pKD46 helper plasmid encoding λ phage *red*, *gam*, and *exo* genes controlled by an arabinose inducible promoter. Recombinant bacteria were selected on LB plates containing chloramphenicol. Following recombination, the chloramphenicol resistance cassette was removed using the flippase encoded on the pCP20 helper plasmid. Correct gene replacement and resistance cassette deletions were confirmed by PCR. Kanamycin resistance was obtained by integrating the *aphT* resistance cassette together with a neutral barcode (WITS13) in *malX* near *ygdA* as described in [57]. Bacteriophage P22 HT/int-mediated transduction was used to transfer mutations into the desired genetic background [58]. All primers used in this study are listed in **S7 Table**.

### Stress resistance analysis by flow cytometry

A total of 10 μl of late exponential phase subcultures were diluted in 90 μl of stress media or vehicle combined with dead staining. Stress media contained either 15% (w/v) sodium cholate (Sigma) in PBS or 100 mM Tris (Sigma) −10 mM EDTA (Sigma) in distilled water. Dead staining was either PI (Invitrogen Waltham, MA) at a final concentration of 30 μg/ml or Sytox blue (Invitrogen) at a final concentration of 10 μM. The mixtures were incubated in 96-well plates for 30 minutes at 37˚C. After incubation, cells were diluted 1/10 in filtered PBS and analyzed by flow cytometry using LSR Fortessa (BD Biosciences Franklin Lakes, NJ) operated with the FACS Diva software (BD Biosciences). Data acquisition was performed until 50,000 events of unstained cells were recorded using excitation with 561-nm laser and band-pass filter 610/20 nm for PI and excitation with 405 nm laser and band-pass filter 450/50 nm for Sytox Blue. The GFP signal was recorded using excitation with 488 nm laser and band-pass filter 512/25 nm and 505LP. Data were processed using FlowJo V10 software (FlowJo Ashland, ORE). Events

were first gated for unstained cells (PI or Sytox blue negative) and further for GFP positive cells when reporter strains were used.

## Membrane potential analysis by flow cytometry and fluorescence microscopy

Late exponential phase subcultures (1 ml) were centrifuged 3 minutes at 9,500 rpm. Supernatants were discarded, and cell pellets were resuspended in 1 ml 10 mM Tris-1 mM EDTA before staining by $DiOC_2(3)$ or PBS before staining by $DiSC_3(5)$. Staining were carried out in 96-well plates in a total volume of 50 µL. In the control wells, CCCP was added at a final concentration of 100 µM in DMSO (Sigma). Cells were incubated for 30 minutes at 37°C. After incubation, $DiOC_2(3)$ (Invitrogen) was added to a final concentration of 30 µM or $DiSC_3(5)$ (Invitrogen) was added to the final concentration of 2 µM. Both dyes were prepared in DMSO. The cells were incubated for 5 minutes at room temperature. Moreover, 20 µL were transferred into 180 µL of 10 mM Tris-1 mM EDTA (for $DiOC_2(3)$ staining) or PBS (for $DiSC_3(5)$ staining) and analyzed by flow cytometry using LSR Fortessa (BD Biosciences) operated by FACS Diva software (BD Biosciences). A total of 50,000 events were recorded using excitation with 488-nm laser and band-pass 542/27 for green fluorescence and excitation with 488-nm laser and band-pass 685/35 for red fluorescence with $DiOC_2(3)$ staining or 640-nm laser and band-pass 670/14 for red fluorescence with $DiSC_3(5)$. Data were processed using FlowJo V10 software (FlowJo).

For fluorescence microscopy imaging, 4 µl of cells were added onto 1% agarose pads directly after the incubation step with $DiOC_2(3)$ or $DiSC_3(5)$. Imaging was performed using an inverted Nikon Ti Eclipse epifluorescence microscope (Nikon, Japan) equipped with a Plan Apo 100× oil immersion objective and a pco.edge 4.2 sCMOS camera. Fluorescence was excited by a SPECTRA X light engine and filtered with a Chroma 84100bs polychroic filter set. FITC setting with cyan laser line (50%, 100 ms exposure), 470/24 nm excitation filter, and 515/30 nm emission filter was used for green fluorescence or with 705/72 nm emission filter for red fluorescence with $DiOC_2(3)$ staining. Cy5 setting with red laser line (50%, 100-ms exposure), 640/30 nm excitation filter and 705/72 nm emission filter was used for red fluorescence with $DiSC_3(5)$.

## Proteomics analysis

**Culture and sorting.** WT SL1344 cells harboring the GFP reporter for HilD regulon expression (fusion P*prgH*::*gfp*) in late exponential phase were collected by centrifugation, resuspended, and diluted in PBS. Samples were sorted according to green fluorescence intensity using a FACS Aria III (BD Biosciences) with scatter and fluorescence channels for GFP, excitation 488 nm, with 514/30nm band-pass with the precision on yield. The flow cytometer was calibrated with CST beads. During the sorting process, the sample and the collection tubes were kept at 4°C or on ice.

After sorting, cells were collected by centrifugation at 12,000 rpm for 10 minutes at 4°C. After each centrifugation step, cells resuspended in the remaining supernatant were transferred to smaller tubes: first 5 ml and then 1.5-ml protein low binding microcentrifuge tubes (Eppendorf Germany). After the final centrifugation step, pellets were stored at −80°C until further processing.

**Sample preparation.** Frozen sorted cell pellets were lysed in 20 µL of lysis buffer (1% sodium deoxycholate (SDC), 10 mM Tris(2-carboxyethyl)phosphin (TCEP), 100 mM Tris, pH = 8.5) using 20 cycles of sonication (30 seconds on, 30 seconds off per cycle) on a Bioruptor (Diagenode SA, Belgium). Following sonication, proteins in the bacterial lysate were heated at

95˚C for 10 minutes. Proteins were then alkylated using 15 mM chloroacetamide at 37˚C for 30 minutes and further digested using sequencing-grade modified trypsin (1/25, w/w, trypsin/ protein; Promega, USA) at 37˚C overnight. After digestion, the samples were acidified with TFA to a final concentration of 1%. Peptides desalting was performed using iST cartridges (PreOmics, Germany) following the manufacturer's instructions. After drying the samples under vacuum, peptides were stored at −20˚C and dissolved in 0.1% aqueous formic acid solution at a concentration of 0.25 mg/ml upon use.

**Mass spectrometry analysis.**    Peptides were subjected to LC–MS analysis using an Orbitrap Fusion Lumos Mass Spectrometer equipped with a nanoelectrospray ion source (both Thermo Fisher Scientific Waltham, MA). Peptide separation was carried out using an EASY nLC-1200 system (Thermo Fisher Scientific) equipped with a RP-HPLC column (75 μm × 36 cm) packed in-house with C18 resin (ReproSil-Pur C18-AQ, 1.9-μm resin; Dr. Maisch, Germany) and a custom-made column heater (60˚C). Peptides were separated using a step-wise linear gradient from 95% solvent A (0.1% formic acid, 99.9% water) and 5% solvent B (80% acetonitrile, 0.1% formic acid, 19.9% water) to 35% solvent B over 45 minutes, to 50% solvent B over 15 minutes, to 95% solvent B over 2 minutes, and 95% solvent B over 18 minutes at a flow rate of 0.2 μl/min.

The mass spectrometer was operated in DDA mode with a cycle time of 3 seconds between master scans. Each master scan was acquired in the Orbitrap at a resolution of 240,000 FWHM (at 200 m/z) and a scan range from 375 to 1,600 m/z followed by MS2 scans of the most intense precursors in the linear ion trap at "Rapid" scan rate with isolation width of the quadrupole set to 1.4 m/z. Maximum ion injection time was set to 50 ms (MS1) and 35 ms (MS2) with an AGC target set to 250% and "standard," respectively. Only peptides with charge state 2 to 5 were included in the analysis. Monoisotopic precursor selection (MIPS) was set to peptide, and the intensity threshold was set to 5e3. Peptides were fragmented by HCD (higher-energy collisional dissociation) with collision energy set to 35%, and one microscan was acquired for each spectrum. The dynamic exclusion duration was set to 30 seconds.

**Protein identification and label-free quantification.**    The acquired raw files were imported into the Progenesis QI software (v2.0, Waters Nonlinear Dynamics United Kingdom), which was used to extract peptide precursor ion intensities across all samples applying the default parameters. The generated mgf files were searched using MASCOT against a decoy database containing normal and reverse sequences of the *Salmonella* Typhimurium (strain SL1344) (UniProt, release date 07.01.2019, in total 10,098 sequences) and commonly observed contaminants generated using the SequenceReverser tool from the MaxQuant software (Version 1.0.13.13). The following search criteria were used: full tryptic specificity was required (cleavage after lysine or arginine residues, unless followed by proline); 3 missed cleavages were allowed; carbamidomethylation (C) was set as fixed modification; oxidation (M) and protein N-terminal acetylation were applied as variable modifications; and mass tolerance of 10 ppm (precursor) and 0.6 Da (fragments). The database search results were filtered using the ion score to set the false discovery rate (FDR) to 1% on the peptide and protein level, respectively, based on the number of reverse protein sequence hits in the datasets.

Quantitative analysis results from label-free quantification were normalized and statistically analyzed using the SafeQuant R package v.2.3.4 (https://github.com/eahrne/SafeQuant (PMID: 27345528)) to obtain protein relative abundances. This analysis included global data normalization by equalizing the total peak/reporter areas across all LC–MS runs, summation of peak areas per protein and LC–MS/MS run, followed by calculation of protein abundance ratios. Only isoform specific peptide ion signals were considered for quantification. The summarized protein expression values were used for statistical testing of between condition differentially abundant proteins. Here, empirical Bayes moderated *t* tests were applied, as

implemented in the R/Bioconductor limma package (http://bioconductor.org/packages/release/bioc/html/limma.html). The resulting per protein and condition comparison *p*-values were adjusted for multiple testing using the Benjamini–Hochberg method.

## Time-lapse microscopy

For HS, 1 mL of the overnight culture was centrifuged and cells resuspended in the same volume of 0.85% KCl (Sigma). Moreover, 70 μl of the suspension was transferred into a 250 μl PCR tube and incubated in a thermocycler (Biometra Germany) 15 minutes at 51+/−0.5˚C. After this, cells were diluted 1/50 in PBS.

For TE treatment, cells in late exponential were diluted 1/10 in 100 mM Tris-10 mM EDTA in ddH$_2$O (treatment) or ddH$_2$O (control) prewarmed at 37˚C and incubated at 37˚C for 30 minutes without shaking. After this, the cells were diluted 1/5 in PBS.

Treated and control cells were inoculated onto a thin matrix of LB agarose attached to a microscope slide. These slides were prepared as follows: a sticky frame (Gene Frame AB-0578) was attached to a standard microscope slide. The resulting cavity was filled with heated LB supplemented with 2% agarose (A4718; Sigma) and covered with a standard microscope slide. After cooling and removal of the cover slide, strips of agarose were removed with the use of a surgical scalpel blade, resulting in 2 rectangles of LB agarose (approximately 5 × 7 mm side), one in the center of each half of the frame. Cells (2 μl) were spotted in the center of the pads, and the frame was sealed with a coverslip (24 × 60 mm). Each slide contained one control and one treated sample.

Microscope slides were examined using an inverted microscope (DeltaVision Core, Cytiva Life Sciences Marlborough, MA, mounted on a motorized Olympus IX71 stand) equipped with an environmental chamber (World Precision Instruments Sarasota, FL) set at 37˚C for up to 16 hours. Images were acquired using a 60× 1.42 NA Plan Apo N objective (Olympus Japan), with the matching 1.524 refractive index immersion oil (Cargille-Sacher Laboratories Cedar Grove, NJ) and a solid-state illumination (Spectra X light engine, Lumencore Beaverton, OR).

Images were recorded with a CoolSNAP HQ2 (Photometrics Tuscon, AZ) at each field every 10 minutes using brightfield (filter "POL", transmittance 32% and exposure time 0.015 seconds) and fluorescence once per hour to prevent phototoxicity (single band-pass emission filter 525/48 nm, transmittance 32%, and exposure time 0.025 seconds). Up to 12 fields were selected per condition and acquired using the microscope control software (DeltaVision SoftWoRx Suite 7.1.0, Cytiva Life Sciences). Images have been analyzed by manual counting using Fiji (ImageJ 1.53c). GFP+ and GFP− cells were identified according to a unique threshold based on fluorescence intensity that was applied on the first picture of every series. On average (+/− standard deviation), 565+/−224, 636+/−248, 493+/−117, and 436+/−118 cells were analyzed per experiment in HS, HS control, TE and TE control conditions, respectively.

## Microfluidic time-lapse microscopy

B04A-03 microfluidic plates for bacteria were used with the CellASIC ONIX2 Microfluidic System to observe the effect of TE treatment in real time at single cell level. Cells in late exponential phase were diluted 1/2 in spent LB medium and loaded into a CellASIC plate using the standard loading sequence protocol. With a constant pressure of 15 kPa, the cells were exposed to spent LB medium for 1 hour before switching to the TE treatment supplemented with PI (30 μg/ml) or Sytox blue (10 μg/ml). Cells trapped in the 0.7-μm height-restricted compartments were imaged with a 2-minute acquisition frame rate on an inverted Nikon Ti Eclipse epifluorescence microscope equipped with a Plan Apo 100x oil immersion objective, a

motorized stage and perfect focus system for multiposition time-lapse imaging, and a pco.edge 4.2 sCMOS camera. Fluorescence was excited by a SPECTRA X light engine and filtered with a Chroma 84100bs polychroic filter set. For GFP fluorescence, FITC settings were used (cyan laser line, 470/24 nm excitation filter, and 515/30 nm emission filter). Cy5 settings (red laser line, 640/30 nm excitation filter, and 705/72 nm emission filter) were used for PI fluorescence, while the CFP settings were used for Sytox blue imaging (blue laser line, 440/20 nm excitation filter, and 475/20 nm emission filter). Postprocessing was done with ImageJ. Fluorescence intensities of individual cells at time points 0, 30, and 60 minutes after start of TE treatment were measured with the point tool set to auto-measure and auto-next slice. Intensity cutoffs were set guided by distribution analysis to 180 for PI and 1,600 for GFP.

## NPN uptake assay

Cells in late exponential phase were centrifuged 3 minutes at 9,500 rpm, and the pellet was washed twice in PBS. NPN uptake was measured in a flat bottom 96-well plate (Corning 3904 Corning, NY) with a final NPN (Sigma) concentration of 10 μM. 2.5 μg/mL Polymyxin B (Sigma) was used as permeabilizing agent in the positive controls. In order to achieve an OD ranging from 0.4 to 0.6, 120 μl of the cell suspension was added to each well on top of 80 μl of PBS (with polymyxin B for the positive controls). NPN was added last, and the measurements were performed immediately.

The fluorescence readout was recorded using a plate reader Synergy H4 Hybrid Reader (BioTek Instruments Winooski, VT) controlled with the Gen5 software. The following excitation/emission wavelengths (in nm) were used: 350/420 for the NPN signal and 491/512 for GFP signal; the bandwidth was 9 nm in all cases. The emission was recorded from the top (Optics: top) with a gain of 100 for GFP and 80 for NPN. OD was measured using 600 nm wavelength. The experiments were performed at 37°C with fast continuous shaking and NPN signal measurements were collected with 1-minute intervals for 1 hour. GFP and OD measurements were taken at the beginning and at the end of the run. We considered the fluorescence values after stabilization of the signal. Thus, one time point was selected per experiment, approximately 45 minutes after the start of the measurements (ranging from 35 to 50 minutes). Wells containing bacteria in DMSO were used to measure the auto-fluorescence of the bacteria. These values were corrected for the OD and subtracted from NPN/OD fluorescence values.

## *In vivo* competitions

Nine- to 12-week-old specified opportunistic pathogen free (SOPF) C57BL/6 mice were pretreated with 25 mg of streptomycin by oral gavage 24 hours prior to infection with *S.*Tm strains to allow robust colonization [59]. Competitors and helper strains were grown separately overnight in LB containing the appropriate antibiotics. Prior to infection, the bacteria were grown to late exponential phase in LB without antibiotics. Cells were washed in PBS, diluted, and mixed to obtain a final inoculum provided to mice by oral gavage. Each 50 μl inoculum contained a total of c.a. $5 \times 10^7$ CFU consisting of the helper strain and 2 competitors at a 100:1:1 ratio. Fecal samples were collected daily, homogenized in 1-ml PBS by bead beating, and bacterial populations were enumerated by selective plating on MacConkey agar containing the appropriate antibiotics. In addition, samples were frozen for determination of the LCN2 concentration. Competitions were allowed to proceed for 3 days p.i. and then the mice were euthanized. To calculate the competitive indexes, population sizes of the 2 competing strains were enumerated by selective plating. The ratio of these 2 values was normalized to the ratio of the 2 competing strains in the corresponding inoculum. To determine the

inflammatory state of the gut, serial dilutions of fecal samples were analyzed using the Mouse LCN2-2/NGAL DuoSet ELISA kit (R&D Systems Minneapolis, MN) according to the manufacturer's instructions.

## Ethics statement

All animal experiments were approved by the legal authorities (Basel-Stadt Kantonales Veterinäramt, license #30480) and follow the 3R guidelines to reduce animal use and suffering to its minimum.

## Statistical analysis

Statistical analysis was conducted using GraphPad Prism 8.0.2.

For Figs 1–3 and S2, S6, S9, and S12 Figs, an important source of noise between repetitions was day-to-day variations in the induction level of the HilD regulon. Such variation is clearly visible in Fig 2D. Small changes in media composition may explain this phenomenon. This generates substantial variability in NPN uptake and stress sensitivity measurements between independent experiments. Therefore, to be able to compare mutants not always studied in parallel, we normalized the datasets using values from WT internal controls always performed in parallel with a given mutant or condition. A value of 1 means value for a given mutant equals value for the WT. After normalization to the WT, outliers were identified using the ROUT method [60] with Q = 1%. Statistical significance of differences between mutants and WT strains in parallel experiments was assessed using the nonnormalized dataset in Wilcoxon matched pairs signed rank tests. The normalized dataset was used in Mann–Whitney tests when comparing mutants or conditions. In S2 and S5 Tables, we followed the **Benjamini–Hochberg procedure** to control the FDR ($\alpha$ = 0.05) [61], meaning that only comparisons yielding *p*-values equal or below the largest *p*-value below its corresponding Benjamini–Hochberg critical value were considered statistically significant.

Violin plots show the empirical distribution of the data, the median, and quartiles. The number of independent biological replicates per condition is indicated below the figures or in figure legends (*n* = x). In experiments where *n* < = 9, single data points are shown on top of the violin plot.

## Supporting information

**S1 Fig. Comparative proteome of *S*.Tm cells sorted according to the expression of P*prgH*::*gfp*.** (**A**) Histogram showing the P*prgH*::*gfp* reporter expression pattern in WT, *ΔhilD*, and *ΔhilE* strains. (**B**) Volcano plot showing differential abundance of proteins in sorted GFP+ (ON) and GFP− (OFF) cells of *S*.Tm WT reporter strain. S1 Table lists all these proteins and their functions and provides details about fold change, statistics, and loci. Source data are provided as a source data file (S1 Data). *S*.Tm, *Salmonella* Typhimurium; WT, wild-type. (EPS)

**S2 Fig. Control of GFP expression from pM965 and NPN uptake in the absence of arabinose (corresponding to Fig 1).** (**A**) Histogram showing the *gfp* expression pattern in strains WT P*prgH*::*gfp* (red) and WT carrying the plasmid pM965 P*rpsM*::*gfp* constitutively expressing *gfp* (blue). (**B, C**) NPN fluorescence corrected by the optical density at 600 nm and normalized to WT pBAD24. (B) Strains harboring pBAD24. (C) Strains harboring p*hilA*. S2 Table lists *p*-values calculated for relevant comparisons. For comparisons against the WT, *p*-values were calculated using the raw data in paired Wilcoxon tests. For comparisons between mutants, *p*-values were calculated using the normalized data in unpaired Mann–Whitney tests. Source

data are provided as a source data file (**S1 Data**). NPN, N-phenyl-1-naphthylamine; WT, wild-type.
(EPS)

**S3 Fig. Growth rate and HS resistance in different genetic backgrounds.** Time-lapse microscopy analysis of WT P*prgH*::*gfp* after HS (50.5˚C, 15 minutes) and untreated control. The cells were observed for 12 hours posttreatment. **(A)** Proportion of cells expressing the HilD regulon (GFP+) from overnight cultures in LB. Triangles: cells used from control experiments, inverted triangles: cells after HS (t = 0). **(B)** Division time of cells expressing or not the HilD regulon from control experiments. **(C, D)** Proportion of cells able to grow in control conditions (C) or after HS (D). **(E)** Superimposed phase contrast and epifluorescence images of representative fields of view in control (upper panels) and HS (lower panels) experiments at t = 0, 3 hours or 7 hours posttreatment. Scale bar corresponds to 5 μm. *p*-Values were calculated using ANOVA and corrected by a Tukey HSD post hoc test. Three independent replicates. Source data are provided as a source data file (**S1 Data**). HS, heat shock; LB, Lysogeny broth; WT, wild-type.
(PDF)

**S4 Fig. Discrimination of cells according to membrane potential revealed by the DiOC$_2$(3) staining. (A)** Flow cytometry plots showing events positioned according to intensities of their red (y-axis) and green fluorescence (x-axis) signals. Unstained WT control allowed delimiting the gate for stained cells. WT cells treated with 100 μm CCCP and 30 μm DiOC$_2$(3) were used to delimit cells with low membrane potential (red) among the stained cells (green). This gating strategy has been used in all independent repetitions to calculate the fraction of cells with low membrane potential in different *S*.Tm populations (**Fig 2**). **(B)** Fluorescence microscopy imaging of cells treated the same as for flow cytometry shows the DiOC$_2$(3) staining of individual cells (all cells green, high membrane potential cells green and red) and the loss of membrane potential after CCCP treatment (no red cells). Source data are provided as a source data file (**S1 Data**). CCCP, carbonyl cyanide 3-chlorophenylhydrazone; DiOC2(3), 3,3′-diethyloxa-carbocyanine iodide; *S*.Tm, *Salmonella* Typhimurium; WT, wild-type.
(EPS)

**S5 Fig. Membrane potential in the absence of stress revealed with DiSC$_3$(5). (A)** Red fluorescence signal from all cells (Total) or cells expressing (GFP+) or not (GFP−) the HilD regulon from WT (left panel) or *ΔhilE* (right panel) strains carrying the reporter P*prgH*::*gfp* (representative experiment). As negative control, 100 μm CCCP was as used to reduce the membrane potential in all cells. **(B)** Mean (left) and median (right) of the red fluorescent signal due to accumulation of DiSC$_3$(5) in WT or *ΔhilE* GFP+ or GFP− populations. *p*-Values were calculated using the paired Wilcoxon test. **(C)** Imaging of DiSC$_3$(5)-stained cells carrying the P*prgH*::*gfp* reporter by fluorescence microscopy. The loss of red fluorescence signal revealed the reduction in membrane potential upon CCCP treatment (lower panels). Source data are provided as a source data file (**S1 Data**). CCCP, carbonyl cyanide 3-chlorophenylhydrazone; DiSC3(5), 3,3′-dipropylthiadicarbocyanine iodide; WT, wild-type.
(EPS)

**S6 Fig. Validation of the cytometry analysis of cells exposed to TE. (A)** Normalized frequency of dead *ΔhilE* reporter cells stained with either Sytox blue or PI after treatment with 100 mM Tris-10 mM EDTA. *n* = x indicates the number of repetitions. **(B)** Normalized frequency of cells stained with either Sytox blue (blue dots) or PI (red dots) after treatment with 100 mM Tris-10 mM EDTA measured by flow cytometry. The graph shows the results for WT, *ΔhilD*, and *ΔhilE* carrying the chromosomal reporter P*prgH*::*gfp* or the plasmidic P*rpsM*::

*gfp* (constitutive GFP expression from pM965). The dataset is normalized by values obtained with the WT reporter strain. For comparisons against the WT, *p*-values were calculated using raw data in paired Wilcoxon tests. For comparisons between mutants or conditions, *p*-values were calculated using normalized data in unpaired Mann–Whitney tests. *n* = 10 repetitions. **(C)** Reduction of the GFP positive fraction (ΔGFP+ in percentage) among WT, *ΔhilD*, or *ΔhilE* cells negative for Sytox blue (blue dots) or PI (red dots)) treated with 100 mM Tris–10 mM EDTA compared to distilled water control. In these control experiments, the GFP was constitutively expressed (P*rpsM*::*gfp*). Significance of the deviation of the median from 0 estimated by Wilcoxon signed rank test. There was no significant loss of GFP + cells after treatment compared to control. *n* = x indicate the number of repetitions. In all panels, *p*-values are indicated within the graph below a bar marking the 2 compared conditions. **(D)** Imaging of WT cells by fluorescence microscopy during exposure to 100 mM Tris-10 mM EDTA (time points 0, 30, and 60 minutes) in the presence of PI or Sytox blue in a CellAsic microfluidic device. Both dyes accumulate within cells exposed to lethal stress, while almost no cells were stained at the beginning of the treatment. Source data are provided as a source data file (**S1 Data**). PI, propidium iodide; TE, Tris-EDTA; WT, wild-type.
(PDF)

**S7 Fig. Single cell level analysis of lethal TE exposure in a microfluidic device (quantitative analysis of S1 Movie). (A)** Total amount of WT P*prgH*::*gfp* cells expressing GFP (ON; green bar) or not (OFF; gray bar), or stained by PI (PI+; red bar) at t0, 30, and 60 minutes of exposure to 100 mM Tris-10 mM EDTA and 30 μg/ml PI in distilled water. **(B)** Proportion of ON, OFF, or PI stained cells among cells that were OFF or ON at t0. **(C, D)** Distribution of fluorescence intensity in cells after 0, 30, and 60 minutes of TE exposure. The dashed lines indicate the cutoff intensities used to bin GFP+ (ON) and GFP− (OFF) cells (C), and PI+ and PI− cells (D). Cutoff intensities were determined according to the distribution at t0 for the GFP signal and at 30 minutes for the PI signal when positive and negative populations were clearly differentiated. Source data are provided as a source data file (**S1 Data**). TE, Tris-EDTA; WT, wild-type.
(EPS)

**S8 Fig. Time-lapse microscopy analysis of cells exposed to lethal concentration of TE and control.** Time-lapse microscopy analysis of WT P*prgH*::*gfp* treated with 100 mM Tris 10 mM EDTA and untreated control. **(A)** Violin plots showing the fraction of cells able to form microcolonies. **(B)** Fraction of GFP+ cells in regrowing and total population after treatment with 100 mM Tris-10 mM EDTA or untreated control. The *p*-values were calculated using the unpaired Mann–Whitney test, *n* = 6 repetitions. Source data are provided as a source data file (**S1 Data**). TE, Tris-EDTA; WT, wild-type.
(EPS)

**S9 Fig. Death after exposure to lethal concentration of TE in the absence of arabinose (corresponding to Fig 2).** Cytometry analysis of cells stained by Sytox blue or PI after exposure to 100 mM Tris–10 mM EDTA. Data normalized by WT pBAD24. **(A)** Strains harboring pBAD24. **(B)** Strains harboring p*hilA*. **S5 Table** lists *p*-values calculated for relevant comparisons. For comparisons against the WT, *p*-values were calculated on the raw data using paired Wilcoxon tests. For comparisons between mutants, *p*-values were calculated using the normalized data and the unpaired Mann–Whitney test. The FDR (α = 0.05) was controlled by the **Benjamini–Hochberg procedure**. Source data are provided as a source data file (**S1 Data**). TE, Tris-EDTA; WT, wild-type.
(EPS)

**S10 Fig. Overexpression of *hilA* decouples the expression of virulence from sensing environmental stress (corresponding to Fig 3). (A)** Proportion of cells expressing GFP from P*prgH*::*gfp* in WT cells harboring pBAD24 or p*hilA*. Growth in presence of 1 mM Arabinose (Ara) induced expression of *hilA* from p*hilA* and subsequent expression of GFP from P*prgH*:: *gfp* in c.a. 90% of the cells independently from pretreatment with sublethal TE concentration (0.04X TE). **(B)** Representative histograms from flow cytometry analysis. Source data are provided as a source data file (S1 Data). TE, Tris-EDTA; WT, wild-type.
(EPS)

**S11 Fig. Competitions 1 and 2, fecal loads during virulent infections (corresponding to Fig 4). (A, B)** CFUs in feces obtained by selective plating for competition 1 (**A**) and 2 (**B**). Dashed lines represent the detection limits, and bars indicate the medians. Source data are provided as a source data file (S1 Data). CFU, colony-forming unit.
(EPS)

**S12 Fig. Competition 2 and 3 in the absence of inflammation. A.** Deleting *invG* does not increase stress resistance. Normalized proportion of cells stained by Sytox or PI after exposure to 100 mM Tris-10 mM EDTA (TE). Mutants are compared to WT controls performed in parallel. For comparisons against the WT, *p*-values were calculated using paired Wilcoxon tests. For comparisons between mutants, *p*-values were calculated using the unpaired Mann–Whitney test on normalized data. **B to C.** Competitions 3 and 4 using avirulent (Avir) Δ*invG* Δ*ssaV* mutants. **C.** Experimental setup. **D.** Competitive indexes calculated from the relative proportions of each competitor in fecal samples. The dashed line represents a 1:1 ratio between HilD+ and HilD− competitors. Values below this line indicate that the HilD− strain outgrew the HilD+ strain. **E.** Intestinal inflammation estimated by measuring LCN2 concentration in the feces. Concentrations above $10^2$ ng/g are usually detected in feces from inflamed gut. Dots correspond to individual mice. Filled dots (black): competition 3 (Avir (HilD+) versus Avir Δ*hilD* (HilD−)); empty dots: competition 4 (Avir Triple Mutant (HilD+) versus Avir Δ*hilD* Triple Mutant (HilD−)). Two independent inocula per condition. **F and G.** CFUs in feces obtained by selective plating for competition 3 (**F**) and 4 (**G**).The *p*-values were generated with 1-way ANOVA and corrected by a Sidak multiple comparisons test. Bars represent the median. Source data are provided as a source data file (S1 Data). CFU, colony-forming unit; LCN2, Lipocalin 2; WT, wild-type.
(EPS)

**S1 Movie. Killing of *S*.Tm by TE analyzed at the single–cell level in a microfluidic device.** WT P*prgH*::*gfp* cells at late exponential phase in LB were loaded into a CellASIC ONIX microfluidic chip and exposed to exhausted LB 60 minutes before starting treatment with 100 mM Tris-10 mM EDTA and 30 μg/ml PI in distilled water. HilD ON cells (green) were prone to die, therefore losing GFP and turning red due to membrane disruption and staining of their DNA by PI. LB, Lysogeny broth; PI, propidium iodide; *S*.Tm, *Salmonella* Typhimurium; TE, Tris-EDTA; WT, wild-type.
(AVI)

**S1 Table. Proteomic analysis of the HilD regulon on sorted *S*.Tm cells.** Cells from WT P*prgH*::*gfp* in late exponential phase were sorted according to GFP fluorescent signal and their respective proteome were analyzed by mass spectrometry and compared. The table lists genes differentially expressed in GFP+ cells compared to GFP− cells (q value <0.05; −0.5>Log2 ratio>0.5). The corresponding volcano plot is presented in S1 Fig. *S*.Tm, *Salmonella* Typhimurium; WT, wild-type.
(XLSX)

**S2 Table. Statistical analysis Fig 1D and 1E and S2 Fig.**
(XLSX)

**S3 Table. Parameters for estimations of death rates.** This table lists the parameters used to calculate the death rates of WT, *hilE*, and *hilD* strains harboring P*prgH*::*gfp* exposed to lethal TE treatment. TE, Tris-EDTA; WT, wild-type.
(XLSX)

**S4 Table. CFU counts after lethal TE treatment.** Control experiments showing that PI or Sytox blue staining of dead cells upon lethal TE exposure correlates to the amount of dead cells deduced from the amount of colonies after plating on LB agar medium. CFU, colony-forming unit; LB, Lysogeny broth; PI, propidium iodide; TE, Tris-EDTA.
(XLSX)

**S5 Table. Statistical analysis (Fig 2H and 2I and S9 Fig).**
(XLSX)

**S6 Table. Strains and plasmids used in this study.**
(XLSX)

**S7 Table. Primers used in this study.**
(XLSX)

**S1 Data. Numerical values underlying all the figures.**
(XLSX)

## Acknowledgments

We would like to acknowledge Stefan Bassler, the FACS core facility, the proteomics core facility, and the imaging core facility of the Biozentrum, Simon van Vliet, Frédéric Goormaghtigh, and Johannes Schneider for technical support as well as all the members of Diard laboratory for scientific input on this project. We would like to thank the Hardt group (ETH Zürich) for sharing strains and plasmids.

## Author Contributions

**Conceptualization:** Abram Aertsen, Médéric Diard.

**Data curation:** Médéric Diard.

**Formal analysis:** Isabelle Hug, Médéric Diard.

**Funding acquisition:** Abram Aertsen, Médéric Diard.

**Investigation:** Malgorzata Sobota, Pilar Natalia Rodilla Ramirez, Alexander Cambré, Andrea Rocker, Julien Mortier, Tiphaine Haas, Delphine Cornillet, Isabelle Hug, Médéric Diard.

**Methodology:** Alexander Cambré, Julien Mortier, Théo Gervais, Dany Chauvin, Isabelle Hug, Thomas Julou, Abram Aertsen, Médéric Diard.

**Project administration:** Médéric Diard.

**Resources:** Abram Aertsen, Médéric Diard.

**Supervision:** Abram Aertsen, Médéric Diard.

**Validation:** Médéric Diard.

**Visualization:** Malgorzata Sobota, Pilar Natalia Rodilla Ramirez, Julien Mortier, Isabelle Hug, Médéric Diard.

**Writing – original draft:** Médéric Diard.

**Writing – review & editing:** Alexander Cambré, Isabelle Hug, Thomas Julou, Abram Aertsen, Médéric Diard.

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
