## [Editor Report · Decision Letter 0]

31 Aug 2021

Dear Dr. Diard, 

Thank you for submitting your manuscript entitled "The expression of virulence increases membrane permeability and sensitivity to envelope stress in Salmonella Typhimurium" for consideration as a Short Reports by PLOS Biology.

Your manuscript has now been evaluated by the PLOS Biology editorial staff and I am writing to let you know that we would like to send your submission out for external peer review.

Please re-submit your manuscript within two working days, i.e. by Sep 02 2021 11:59PM.

Kind regards,

Paula 

---

Paula Jauregui, PhD

Associate Editor

PLOS Biology

---

## [Decision Letter · Decision Letter 1]

2 Nov 2021

Dear Dr. Diard,

Thank you for submitting your manuscript "The expression of virulence increases membrane permeability and sensitivity to envelope stress in Salmonella Typhimurium" for consideration as a Short Reports at PLOS Biology. Your manuscript has been evaluated by the PLOS Biology editors, an Academic Editor with relevant expertise, and by several independent reviewers.

In light of the reviews (below), we will not be able to accept the current version of the manuscript, but we would welcome re-submission of a much-revised version that takes into account the reviewers' comments. We cannot make any decision about publication until we have seen the revised manuscript and your response to the reviewers' comments. Your revised manuscript is also likely to be sent for further evaluation by the reviewers.

In particular, reviewer #1 thinks that you don’t show evidence that the Pprg-gfp construct is directly activated by HilD, questions your presentation of the data, thinks you need to show representative images of the cell populations for all experiments, points out that over-expressing SPI-1 regulators on a plasmid is not appropriate for studying regulation in response to environmental stress, has several questions about the mouse experiment, and wants you to add more detail in the method section. Reviewer #2 thinks that you should add more detail in the abstract, points out several text changes for clarity, labels from figures, missing spaces, and that the strain table is lacking complete references. Reviewer #3 suggests another mouse experiment to test whether hilD mutants in the wild-type and triple mutant background no longer have a different phenotype in absence of intestinal inflammation, thinks that you should add additional inflammation data in the mouse experiment, and asks how important motility for intestinal colonization is.

We expect to receive your revised manuscript within 3 months. 

**IMPORTANT - SUBMITTING YOUR REVISION**

*Re-submission Checklist*

*Published Peer Review*

*PLOS Data Policy*

*Blot and Gel Data Policy*

Sincerely,

Paula 

---

Paula Jauregui, PhD

Associate Editor

PLOS Biology

REVIEWS:

Reviewer #1: Salmonella virulence, outer membrane and single cell.

Reviewer #2: Salmonella-host interactions.

Reviewer #3: Bacteria-host interactions

Reviewer #1: The expression of virulence increases membrane permeability and sensitivity to envelope stress in ST by Sobota et al describes experiments that address the question of fitness cost to a pathogen for expressing virulence genes. The model is Salmonella Typhimurium, and the authors explore how expression of HilD and presumably HilD-regulated genes is stressful. It is an interesting premise, but the manuscript is seriously flawed, not only in its writing, but in the presentation of the data. The use of ratios and violin plots makes it very difficult to discern what is going on. Perhaps binning the results and their distribution would give a better overall picture of what is happening in the overall population. Furthermore, representative images need to be shown associated with each experiment.

1. The authors employ a Pprg-gfp construct as a reporter for HilD, but they never show evidence that it is directly activated by HilD. If this has been previously shown, it should be stated explicitly and referenced. From my reading of the literature, HilD is not directly regulating prgH (ref 8), which in my view is problematic for their analysis. 

2. Why is the data always presented as a function of OD? It makes it hard to understand what is actually being plotted. If every sample is at the same OD, then why don't the authors present the data simply as relative fluorescence units? If all the samples are n=7, this does not need to be presented on the graph. It should be included in the legend. What do the numbers in the figure indicate? I can't find this in the legend. What was the level of autofluorescence and how was it corrected? P values for figures should NEVER be listed separately in SI. In Figure 1B, why is there so much more variation in the �hilE strain compared to the GFP?? Using a control that expresses GFP on a plasmid does not evenly compare copy numbers. I would like to see an image of the membranes of the two populations, those expressing HilD and those that show reduced NPN partitioning. It isn't clear how reduced uptake of NPN is correlated with a reduction in membrane integrity, and the authors have not made this link clearly. For all experiments, representative images of the cell populations should be shown. 

3. Figure 1D/E are described before Fig. 1C, which was never described. 

4. The results section begins with all data that is in Supplementary information. This does not make the manuscript easy to read or straight forward. The construction of the manuscript and its presentation style is difficult. The tables in Supplemental information are not well annotated. The authors need to distill the essential features of the table, put it in the regular manuscript and then have the full table in SI. 

5. The way the data are plotted seem to obscure the single cell differences. Are there two distinct populations that are responding? And if so, why not separate them and analyze them separately? The differences would be more compelling.

6. Over-expressing SPI-1 regulators on a plasmid is not appropriate for studying regulation in response to environmental stress, especially when expression is not homogeneous.

7. The authors try to make the correlation between NPN sensitivity and TE sensitivity with respect to �SPI-1 and �flg strains, but the results do not compare when looking at Sytox or PI staining of dead cells (Fig. 2H-I and Fig. 1E). This would seem to negate their hypothesis that pathogenesis creates membrane stress and sensitivity to environmental stress.

8. In the mouse experiment, the triple mutant does better, but by day 3, the difference is very small. One wonders what would have happened on day 4. Why were differing numbers of mice (only 2 WT and 3 hilD- ?) used? And there is very little difference in inflammation, why? The legend is poorly explained and so the figure is difficult to interpret. 

9. The method section is very poorly written and lacking in detail. 

Minor:

1.Lines 39-41 The authors imply that the relationship between environmental stimuli and the regulation of virulence gene expression is not understood. This statement is way overblown. Lines 51-52 the authors state: However, the fitness-cost of virulence at the single-cell level and the facultative intra-cellular lifestyle of S .Tm strongly constrain the expression of the HilD regulon. This sentence is misleading, what turns off HilD is not the fitness cost, it is the acidic environment of the vacuole, that also directly stimulates the SsrB regulon and SPI-2 gene expression (which should be associated with a different fitness cost). This is implied but not well stated in lines 56-68. References here are out of date and not comprehensive. In the subsequent paragraph, the other key two-component systems (EnvZ-OmpR and SsrA-SsrB) systems are completely ignored. 

2. Embedding the figure legends in the text was a super pain to read and disrupts the flow. 

3. The Introduction reads more like a grant proposal than a manuscript submission and raises a number of questions, but doesn't clearly describe their hypothesis and how they will test it. 

4. Line 263 is enable supposed to be unable?

5. Fig. 2B is too small. 

6. As an example of writing, for example, instead of stating "lesser resistance to stress" say increased sensitivity to stress-the direct is more clear than the indirect. Examples of this kind of writing are present throughout the ms.

7. What does 1ml of 4h day cultures mean (line 471)?

Reviewer #2: This is an interesting manuscript dissecting the envelope stress induced in Salmonella by the coordinated expression of primarily the SPI1 T3SS and flagellar apparatus. The experiments are thorough and well controlled. Interestingly, they distinguish between envelope stress and a general growth defect induced by HilD expression. My comments focus on presentation and grammar. 

Line 21. "…could allow the identification and modulation of ecological factors…"

The abstract could provide more detail of the results, explicitly the points made in the summary paragraph above.

Line 85. Delete "ectopic". More importantly, state where this fusion is the chromosome, emphasizing that it does not disrupt SPI1 T3SS expression. 

Line 114. Replace "it" with "GFP". As it is now written, the reader is not sure if "it" refers to GFP or HilD. 

Line 144. In several figure legends, you refer to "day cultures". I do not know what that means. 

Line 178. You use the term "polarization" to mean maintenance of the proton motive force. Be more explicit.

Line 209. You can delete the line "Thoroughly filtered media…"

Line 263. I think you mean "able" not "enable". 

Figs 2D, 3A, and 3C. I found the labels on the y axis to be very confusing. I thought you were measuring the fraction of GFP+ cells that were alive. I finally figured out it was the fraction of GFP+ cells. Maybe just remove the word "alive". 

Fig 3 C and D. It might be useful to move the "sub-lethal" pluses and minuses above the names. With the slanted names, it is a little difficult to know if the plus or minus refers to the start of the name immediately above it or the data directly above it. It's confusing to the eye. 

You are missing spaces in many places: before parentheses and between numbers and units. 2mg versus 2 mg, for example. 

Your strain table is lacking the complete references.

Reviewer #3: With this paper the authors extend our understanding of fitness trade-offs of pathogenic bacterial virulence expression on the example of Salmonella Typhimurium. The senior author's previous work on this topic has progressed the field considerably, and the key findings in this work in Salmonella are likely relevant also for other pathogens. In Salmonella the HilD regulon regulates many virulence factors and other genes, including the expression of the SPI1 type 3 secretion system (T3SS-1), SPI4, and the flagella system that are all key virulence factors mediating intestinal inflammatory pathology which modifies the pathogens' intestinal niche to its own relative advantage. However, virulence expression regulated by HilD is costly and drastically slowing growth of virulence expressing bacteria, threatening the stability of virulence in Salmonella by selection of avirulent mutants. As shown in the senior author's elegant previous work, this is prevented by the bimodal expression of the HilD regulon, which counters the out-competition of the pathogen by genetically avirulent "cheater" mutants.

In the present paper, inspired by the fact that several key virulence factors, T3SS-1, flagella apparatus, and SPI4 type 1 secretion system under HilD control are major membrane complexes that may impose cell envelope stress, the authors elucidate the contribution of envelope stress to the cost of virulence expression. Single cell-based analyses provide evidence that the HilD controlled expression of virulence inflicts envelope stress (manifesting in increased outer membrane permeability, decreased heat hock survival, increased membrane depolarization and reduced viability in response to EDTA treatment stress) compared to HilD-deficient mutants. Triple deficiency for SPI1 type 3 secretion apparatus, flagella and SPI4 rescued this stress phenotype, whereas overexpression of T3SS-1 and SPI4 exacerbated it. 

Sublethal envelope stress also downregulated HilD regulon induction (with was known before) and consequently increases stress resistance to subsequent lethal envelope stress. 

A very interesting novel finding is that by triple-deficiency for T3SS-1, flagella, and SPI-4 in a HilD proficient genetic background, the envelope stress-related cost of virulence expression can be uncoupled from the slow-growth phenotype of the HilD regulon expressing Salmonella sub-population. The mechanistic nature of the slow growth-related cost of virulence expression remains in the dark. However, the authors conclude the study with an elegant competitive colonization experiment in the streptomycin-pretreated mouse model, showing that the envelope stress related cost of virulence expression does contribute to the growth disadvantage of virulence expressing Salmonella specifically in stressful environment of the inflamed gut. 

This work is of very high quality. The manuscript is very well written and referenced. The introduction provides the relevant background and leads over to the main hypothesis in a easily understandable manner. The in-vitro experiments are very well designed, very well controlled and very carefully interpreted and adequately support the authors' conclusions. In the dioscussion section, the findings are rigorously and critically discussed and put in context with the current literature. 

The only point I would regard worthwhile addressing with additional experimentation regards the final mouse experiment:

1. If the authors' conclusions were correct, hilD mutants in the wild-type and triple mutant background should no longer have a different phenotype in absence of intestinal inflammation. An additional animal experiment could therefore be tried to test this prediction, using an avirulent mutant as helper strain, and the use competing test strains in a SipBC translocon deficient mutant background that is intestinally avirulent but still able to express a fully assembled T3SS-1 apparatus. (I understand that the required additional strain construction may not be trivial, and the authors may be able to provide an alternative, perhaps even better approach to address this point.)

2. Since it is important here that the degree of inflammation is comparable between the two animal groups, the inclusion of additional inflammation data, such as cecal histology data, would be an improvement. Lipocalin-2 quantification is a proven quantitative readout but can be variable between experiments. The histological picture on day 1 may even more convincingly support the claim that the two mice labelled in red are indeed particularly severely inflamed.

3. Minor comment: triple mutant and wild-type background differ in motility. How important is motility for intestinal colonization?

---

## [Editor Report · Decision Letter 2]

23 Feb 2022

Dear Dr Diard,

Thank you for submitting your manuscript "The expression of virulence increases membrane permeability and sensitivity to envelope stress in Salmonella Typhimurium" for consideration as a Short Reports by PLOS Biology. I have now discussed your new version with the Academic Editor and am pleased to let you know that we will probably accept this manuscript for publication, provided you satisfactorily address the following data and other policy-related requests:

1) Title. We would like to suggest a minor modification: "The expression of virulence genes increases membrane permeability and sensitivity to envelope stress in Salmonella Typhimurium".

2) Data: You may be aware of the PLOS Data Policy, which requires that all data be made available without restriction: http://journals.plos.org/plosbiology/s/data-availability. For more information, please also see this editorial: http://dx.doi.org/10.1371/journal.pbio.1001797

Note that we do not require all raw data. Rather, we ask for all individual quantitative observations that underlie the data summarized in the figures and results of your paper. For an example see here: http://www.plosbiology.org/article/info%3Adoi%2F10.1371%2Fjournal.pbio.1001908#s5

These data can be made available in one of the following forms:

I) Supplementary files (e.g., excel). Please ensure that all data files are uploaded as 'Supporting Information' and are invariably referred to (in the manuscript, figure legends, and the Description field when uploading your files) using the following format verbatim: S1 Data, S2 Data, etc. Multiple panels of a single or even several figures can be included as multiple sheets in one excel file that is saved using exactly the following convention: S1_Data.xlsx (using an underscore).

II) Deposition in a publicly available repository. Please also provide the accession code or a reviewer link so that we may view your data before publication.

Regardless of the method selected, please ensure that you provide the individual numerical values that underlie the summary data displayed in the following figure panels: Figures 1 ABCDE, 2 ACDEFGHI, 3 ABCD, 4 BC, S1 B, S2 BC, S3 ABCD, S5 B, S6 ABC, S7 ABCD, S8 AB, S9 AB, S10 AB, S11 AB, S12 ACDEF.

2.1) Please also ensure that each figure legend in your manuscript includes information on where the underlying data can be found and that your supplemental data file/s has/have a legend.

2.2) Please ensure that your Data Statement in the submission system accurately describes where your data can be found.

We expect to receive your revised manuscript within two weeks. 

*Published Peer Review History*

*Early Version*

Sincerely,

Dario

Dario Ummarino, PhD

Senior Editor

PLOS Biology

dummarino@plos.org

---

## [Editor Report · Decision Letter 3]

17 Mar 2022

Dear Dr Diard,

On behalf of my colleagues and the Academic Editor, Matt Waldor, I am pleased to say that we can in principle accept your Short Reports "The expression of virulence genes increases membrane permeability and sensitivity to envelope stress in Salmonella Typhimurium" for publication in PLOS Biology, provided you address any remaining formatting and reporting issues. These will be detailed in an email that will follow this letter and that you will usually receive within 2-3 business days, during which time no action is required from you. Please note that we will not be able to formally accept your manuscript and schedule it for publication until you have completed any requested changes.

IMPORTANT: Many thanks for providing the underlying data supporting your main figures and supplementary ones. I've asked my colleagues to also request that you include the information about data location in all your supplementary figure legends (e.g. "The data underlying this Figure are available at XXX").

PRESS

Sincerely, 

Dario

Dario Ummarino, PhD 

Senior Editor 

PLOS Biology

dummarino@plos.org